# A Universal Growth Rate for
# Learning with Smooth Surrogate Losses

**Anqi Mao**
Courant Institute
New York, NY 10012
aqmao@cims.nyu.edu

**Mehryar Mohri**
Google Research & CIMS
New York, NY 10011
mohri@google.com

**Yutao Zhong**
Courant Institute
New York, NY 10012
yutao@cims.nyu.edu

## Abstract

This paper presents a comprehensive analysis of the growth rate of $\mathcal{H}$-consistency bounds (and excess error bounds) for various surrogate losses used in classification. We prove a square-root growth rate near zero for smooth margin-based surrogate losses in binary classification, providing both upper and lower bounds under mild assumptions. This result also translates to excess error bounds. Our lower bound requires weaker conditions than those in previous work for excess error bounds, and our upper bound is entirely novel. Moreover, we extend this analysis to multi-class classification with a series of novel results, demonstrating a universal square-root growth rate for smooth *comp-sum* and *constrained losses*, covering common choices for training neural networks in multi-class classification. Given this universal rate, we turn to the question of choosing among different surrogate losses. We first examine how $\mathcal{H}$-consistency bounds vary across surrogates based on the number of classes. Next, ignoring constants and focusing on behavior near zero, we identify *minimizability gaps* as the key differentiating factor in these bounds. Thus, we thoroughly analyze these gaps, to guide surrogate loss selection, covering: comparisons across different comp-sum losses, conditions where gaps become zero, and general conditions leading to small gaps. Additionally, we demonstrate the key role of minimizability gaps in comparing excess error bounds and $\mathcal{H}$-consistency bounds.

## 1 Introduction

Learning algorithms frequently optimize surrogate loss functions like the logistic loss, in lieu of the task's true objective, commonly the zero-one loss. This is necessary when the original loss function is computationally intractable to optimize or lacks essential mathematical properties such as differentiability. But, what guarantees can we rely on when minimizing a surrogate loss? This is a fundamental question with significant implications for learning.

The related property of Bayes-consistency of surrogate losses has been extensively studied in the context of binary classification. Zhang [2004a], Bartlett et al. [2006] and Steinwart [2007] established Bayes-consistency for various convex loss functions, including margin-based surrogates. They also introduced excess error bounds (or surrogate regret bounds) for margin-based surrogates. Reid and Williamson [2009] extended these results to proper losses in binary classification.

The Bayes-consistency of several surrogate loss function families in the context of multi-class classification has also been studied by Zhang [2004b] and Tewari and Bartlett [2007]. Zhang [2004b] established a series of results for various multi-class classification formulations, including negative results for multi-class hinge loss functions [Crammer and Singer, 2001], as well as positive results for the sum exponential loss [Weston and Watkins, 1999, Awasthi et al., 2022a], the (multinomial) logistic loss [Verhulst, 1838, 1845, Berkson, 1944, 1951], and the constrained losses [Lee et al., 2004].

38th Conference on Neural Information Processing Systems (NeurIPS 2024).

Later, Tewari and Bartlett [2007] adopted a different geometric method to analyze Bayes-consistency, yielding similar results for these loss function families. Steinwart [2007] developed general tools to characterize Bayes consistency for both binary and multi-class classification. Additionally, excess error bounds have been derived by Pires et al. [2013] for a family of constrained losses and by Duchi et al. [2018] for loss functions related to generalized entropies.

For a surrogate loss $\ell$, an excess error bound holds for any predictor $h$ and has the form $\mathcal{E}_{\ell_{0-1}}(h) - \mathcal{E}^*_{\ell_{0-1}} \leq \Psi(\mathcal{E}_\ell(h) - \mathcal{E}^*_\ell)$, where $\mathcal{E}_{\ell_{0-1}}(h)$ and $\mathcal{E}_\ell(h)$ represent the expected losses of $h$ for the zero-one loss and surrogate loss respectively, $\mathcal{E}^*_{\ell_{0-1}}$ and $\mathcal{E}^*_\ell$ the Bayes errors for the zero-one and surrogate loss respectively, and $\Psi$ a non-decreasing function. The *growth rate* of excess error bounds, that is the behavior of function $\Psi$ near zero, has gained attention in recent research [Mahdavi et al., 2014, Zhang et al., 2021, Frongillo and Waggoner, 2021, Bao, 2023]. Mahdavi et al. [2014] examined the growth rate for *smoothed hinge losses* in binary classification, demonstrating that smoother losses result in worse growth rates. The optimal rate is achieved with the standard hinge loss, which exhibits linear growth. Zhang et al. [2021] tied the growth rate of excess error bounds in binary classification to two properties of the surrogate loss function: consistency intensity and conductivity. These metrics enable comparisons of growth rates across different surrogates. This prompts a natural question: can we establish rigorous lower and upper bounds for excess error growth rates under specific regularity conditions?

Frongillo and Waggoner [2021] pioneered research on this question in binary classification settings. They established a critical square-root lower bound for excess error bounds when a surrogate loss is locally strongly convex and has a locally Lipschitz gradient. Additionally, they demonstrated a linear excess error bound for Bayes-consistent polyhedral loss functions (convex and piecewise-linear) [Finocchiaro et al., 2019] (see also [Lapin et al., 2016, Ramaswamy et al., 2018, Yu and Blaschko, 2018, Yang and Koyejo, 2020]). More recently, Bao [2023] complemented these results by showing that proper losses associated with Shannon entropy, exponential entropy, spherical entropy, squared $\alpha$-norm entropies and $\alpha$-polynomial entropies, with $\alpha > 1$, also exhibit a square-root lower bound for excess error bounds relative to the $\ell_1$-distance.

However, while Bayes-consistency and excess error bounds are valuable, they are not sufficiently informative, as they are established for the family of all measurable functions and disregard the crucial role played by restricted hypothesis sets in learning. As pointed out by Long and Servedio [2013], in some cases, minimizing Bayes-consistent losses can result in constant expected error, while minimizing inconsistent losses can yield an expected loss approaching zero. To address this limitation, the authors introduced the concept of *realizable $\mathcal{H}$-consistency*, further explored by Kuznetsov et al. [2014] and Zhang and Agarwal [2020]. Nonetheless, these guarantees are only asymptotic and rely on a strong realizability assumption that typically does not hold in practice.

Recent research by Awasthi, Mao, Mohri, and Zhong [2022b,a] and Mao, Mohri, and Zhong [2023f,c,e,b] has instead introduced and analyzed $\mathcal{H}$-consistency bounds. These bounds are more informative than Bayes-consistency since they are hypothesis set-specific and non-asymptotic. Their work covers broad families of surrogate losses in binary classication, multi-class classification, structured prediction, and abstention [Mao, Mohri, Mohri, and Zhong, 2023a]. Crucially, they provide upper bounds on the *estimation error* of the target loss, for example, the zero-one loss in classification, that hold for any predictor $h$ within a hypothesis set $\mathcal{H}$. These bounds relate this estimation error to the surrogate loss estimation error.

Their general form is: $\mathcal{E}_{\ell_{0-1}}(h) - \mathcal{E}^*_{\ell_{0-1}}(\mathcal{H}) + \mathcal{M}_{\ell_{0-1}}(\mathcal{H}) \leq \Gamma(\mathcal{E}_\ell(h) - \mathcal{E}^*_\ell(\mathcal{H}) + \mathcal{M}_\ell(\mathcal{H}))$, where $\mathcal{E}^*_{\ell_{0-1}}(\mathcal{H})$ and $\mathcal{E}^*_\ell(\mathcal{H})$ represent the best-in-class expected losses for the zero-one and surrogate loss respectively, $\Gamma$ is a non-negative concave function and $\mathcal{M}_{\ell_{0-1}}(\mathcal{H})$ and $\mathcal{M}_\ell(\mathcal{H})$ are *minimizability gaps*. The exact definition of these gaps will be detailed later. For now, let us mention that they are non-negative quantities, upper-bounded by the approximation error of their respective loss functions. $\mathcal{H}$-consistency bounds subsume excess error bounds as a special case when the hypothesis set is expanded to include all measurable functions, in which case the minimizability gaps vanish. More generally, an $\mathcal{H}$-consistency bound with a $\Gamma$ function implies $\mathcal{E}_{\ell_{0-1}}(h) - \mathcal{E}^*_{\ell_{0-1}}(\mathcal{H}) + \mathcal{M}_{\ell_{0-1}}(\mathcal{H}) \leq \Gamma(\mathcal{E}_\ell(h) - \mathcal{E}^*_\ell(\mathcal{H})) + \Gamma(\mathcal{M}_\ell(\mathcal{H}))$ since a concave function $\Gamma$ with $\Gamma(0) \geq 0$ is sub-additive over $\mathbb{R}_+$. Thus, when the surrogate estimation loss $\mathcal{E}_\ell(h) - \mathcal{E}^*_\ell(\mathcal{H})$ is minimized to $\epsilon$, the zero-one estimation error $\mathcal{E}_{\ell_{0-1}}(h) - \mathcal{E}^*_{\ell_{0-1}}(\mathcal{H})$ is bounded by $\Gamma(\epsilon) + \Gamma(\mathcal{M}_\ell(\mathcal{H})) - \mathcal{M}_{\ell_{0-1}}(\mathcal{H})$. Can we characterize the growth rate of $\mathcal{H}$-consistency bounds, that is how quickly the functions $\Gamma$ increase near zero?

**Our results**. This paper presents a comprehensive analysis of the growth rate of $\mathcal{H}$-consistency bounds for all margin-based surrogate losses in binary classification, as well as for *comp-sum losses* and *constrained losses* in multi-class classification. We establish a square-root growth rate near zero for margin-based surrogate losses $\ell$ defined by $\ell(h, x, y) = \Phi(-yh(x))$, assuming only that $\Phi$ is convex and twice continuously differentiable with $\Phi'(0) > 0$ and $\Phi''(0) > 0$ (Section 4). This includes both upper and lower bounds (Theorem 4.2). These results directly apply to excess error bounds as well. Importantly, our lower bound requires weaker conditions than [Frongillo and Waggoner, 2021, Theorem 4], and our upper bound is entirely novel. This work demonstrates that the $\mathcal{H}$-consistency bound growth rate for these loss functions is precisely square-root, refining the "at least square-root" finding of these authors (for excess error bounds). It is known that polyhedral losses admit a linear grow rate [Frongillo and Waggoner, 2021]. Thus, a striking dichotomy emerges that reflects previous observations by these authors: $\mathcal{H}$-consistency bounds for polyhedral losses exhibit a linear growth rate in binary classification, while they follow a square-root rate for smooth loss functions.

Moreover, we significantly extend our findings to key multi-class surrogate loss families, including *comp-sum losses* [Mao et al., 2023f] (e.g., logistic loss or cross-entropy with softmax [Berkson, 1944], sum-losses [Weston and Watkins, 1999], generalized cross entropy loss [Zhang and Sabuncu, 2018]), and *constrained losses* [Lee et al., 2004, Awasthi et al., 2022a] (Section 5). In Section 5.1, we prove that the growth rate of $\mathcal{H}$-consistency bounds for comp-sum losses is exactly square-root. This applies when the auxiliary function $\Phi$ they are based upon is convex and twice continuously differentiable with $\Phi'(u) < 0$ and $\Phi''(u) > 0$ for all $u$ in $(0, \frac{1}{2}]$. These conditions hold for all common loss functions used in practice. Further, in Section 5.2, we demonstrate that the square-root growth rate also extends to $\mathcal{H}$-consistency bounds for constrained losses. This requires the auxiliary function $\Phi$ to be convex and twice continuously differentiable with $\Phi'(u) > 0$ and $\Phi''(u) > 0$ for any $u \geq 0$, alongside an additional technical condition. These are satisfied by all constrained losses typically encountered in practice.

These results reveal a universal square-root growth rate for smooth surrogate losses, the predominant choice in neural network training (over polyhedral losses) for both binary and multi-class classification in applications. Given this universal growth rate, how do we choose between different surrogate losses? Section 6 addresses this question in detail. To start, we examine how $\mathcal{H}$-consistency bounds vary across surrogates based on the number of classes. Then, focusing on behavior near zero (ignoring constants), we isolate minimizability gaps as the key differentiating factor in these bounds. These gaps depend solely on the chosen surrogate loss and hypothesis set. We provide a detailed analysis of minimizability gaps, covering: comparisons across different comp-sum losses, conditions where gaps become zero, and general conditions leading to small gaps. These findings help guide surrogate loss selection. Additionally, we demonstrate the key role of minimizability gaps in comparing excess error bounds and $\mathcal{H}$-consistency bounds (Appendix F). Importantly, combining $\mathcal{H}$-consistency bounds with surrogate loss Rademacher complexity bounds allows us to derive zero-one loss (estimation) learning bounds for surrogate loss minimizers (Appendix O).

For a more comprehensive discussion of related work, please refer to Appendix A. We start with the introduction of necessary concepts and definitions.

## 2 Preliminaries

**Notation and definitions.** We denote the input space by $\mathcal{X}$ and the label space by $\mathcal{Y}$, a finite set of cardinality $n$ with elements $\{1, \ldots, n\}$. $\mathcal{D}$ denotes a distribution over $\mathcal{X} \times \mathcal{Y}$.

We write $\mathcal{H}_{\text{all}}$ to denote the family of all real-valued measurable functions defined over $\mathcal{X} \times \mathcal{Y}$ and denote by $\mathcal{H}$ a subset, $\mathcal{H} \subseteq \mathcal{H}_{\text{all}}$. The label assigned by $h \in \mathcal{H}$ to an input $x \in \mathcal{X}$ is denoted by $\mathsf{h}(x)$ and defined by $\mathsf{h}(x) = \operatorname{argmax}_{y \in \mathcal{Y}} h(x, y)$, with an arbitrary but fixed deterministic strategy used for breaking the ties. For simplicity, we fix that strategy to be the one selecting the label with the highest index under the natural ordering of labels.

We will consider general loss functions $\ell \colon \mathcal{H} \times \mathcal{X} \times \mathcal{Y} \to \mathbb{R}_+$. For many loss functions used in practice, the loss value at $(x, y)$, $\ell(h, x, y)$, only depends on the value $h$ takes at $x$ and not on its values on other points. That is, there exists a measurable function $\hat{\ell} \colon \mathbb{R}^n \times \mathcal{Y} \to \mathbb{R}_+$ such that $\ell(h, x, y) = \hat{\ell}(h(x), y)$, where $h(x) = [h(x, 1), \ldots, h(x, n)]$ is the score vector of the predictor $h$. We will then say that $\ell$ is a *pointwise loss function*. We denote by $\mathcal{E}_\ell(h)$ the generalization error or expected loss of a hypothesis $h \in \mathcal{H}$ and by $\mathcal{E}_\ell^*(\mathcal{H})$ the *best-in class error*: $\mathcal{E}_\ell(h) =$

$\mathbb{E}_{(x,y)\sim\mathcal{D}}[\ell(h,x,y)]$, $\mathcal{E}_\ell^*(\mathcal{H}) = \inf_{h\in\mathcal{H}} \mathcal{E}_\ell(h)$. $\mathcal{E}_\ell^*(\mathcal{H}_{\text{all}})$ is also known as the *Bayes error*. We write $\mathsf{p}(y\mid x) = \mathcal{D}(Y = y \mid X = x)$ to denote the conditional probability of $Y = y$ given $X = x$ and $p(x) = (\mathsf{p}(1\mid x),\ldots,\mathsf{p}(n\mid x))$ for the conditional probability vector for any $x \in \mathcal{X}$. We denote by $\mathcal{C}_\ell(h,x)$ the *conditional error* of $h \in \mathcal{H}$ at a point $x \in \mathcal{X}$ and by $\mathcal{C}_\ell^*(\mathcal{H},x)$ the *best-in-class conditional error*: $\mathcal{C}_\ell(h,x) = \mathbb{E}_y[\ell(h,x,y)\mid x] = \sum_{y\in\mathcal{Y}} \mathsf{p}(y|x)\,\ell(h,x,y)$, $\mathcal{C}_\ell^*(\mathcal{H},x) = \inf_{h\in\mathcal{H}} \mathcal{C}_\ell(h,x)$, and use the shorthand $\Delta\mathcal{C}_{\ell,\mathcal{H}}(h,x) = \mathcal{C}_\ell(h,x) - \mathcal{C}_\ell^*(\mathcal{H},x)$ for the *calibration gap* or *conditional regret* for $\ell$. The generalization error of $h$ can be written as $\mathcal{E}_\ell(h) = \mathbb{E}_x[\mathcal{C}_\ell(h,x)]$. For convenience, we also define, for any vector $p = (p_1,\ldots,p_n) \in \Delta^n$, where $\Delta^n$ is the probability simplex of $\mathbb{R}^n$, $\mathcal{C}_\ell(h,x,p) = \sum_{y\in\mathcal{Y}} p_y\,\ell(h,x,y)$, $\mathcal{C}_{\ell,\mathcal{H}}^*(x,p) = \inf_{h\in\mathcal{H}} \mathcal{C}_\ell(h,x,p)$ and $\Delta\mathcal{C}_{\ell,\mathcal{H}}(h,x,p) = \mathcal{C}_\ell(h,x,p) - \mathcal{C}_{\ell,\mathcal{H}}^*(x,p)$. Thus, we have $\Delta\mathcal{C}_{\ell,\mathcal{H}}(h,x,p(x)) = \Delta\mathcal{C}_{\ell,\mathcal{H}}(h,x)$.

We will study the properties of a surrogate loss function $\ell_1$ for a target loss function $\ell_2$. In multi-class classification, $\ell_2$ is typically the zero-one multi-class classification loss function $\ell_{0-1}$ defined by $\ell_{0-1}(h,x,y) = 1_{h(x)\neq y}$. Some surrogate loss functions $\ell_1$ include the max losses [Crammer and Singer, 2001], comp-sum losses [Mao et al., 2023f] and constrained losses [Lee et al., 2004].

**Binary classification.** The definitions just presented were given for the general multi-class classification setting. In the special case of binary classification (two classes), the standard formulation and definitions are slightly different. For convenience, the label space is typically defined as $\mathcal{Y} = \{-1, +1\}$. Instead of two scoring functions, one for each label, a single real-valued function is used whose sign determines the predicted class. Thus, here, a hypothesis set $\mathcal{H}$ is a family of measurable real-valued functions defined over $\mathcal{X}$ and $\mathcal{H}_{\text{all}}$ is the family of all such functions. $\ell$ is *pointwise* if there exists a measurable function $\hat{\ell}\colon\mathbb{R}\times\mathcal{Y}\to\mathbb{R}_+$ such that $\ell(h,x,y) = \hat{\ell}(h(x),y)$. The target loss function is typically the binary loss $\ell_{0-1}$, defined by $\ell_{0-1}(h,x,y) = 1_{\operatorname{sign}(h(x))\neq y}$, where $\operatorname{sign}(h(x)) = 1_{h(x)\geq 0} - 1_{h(x)<0}$. Some widely used surrogate losses $\ell_1$ for $\ell_{0-1}$ are margin-based losses, which are defined by $\ell_1(h,x,y) = \Phi(-yh(x))$, for some non-decreasing convex function $\Phi\colon\mathbb{R}\to\mathbb{R}_+$. Instead of two conditional probabilities, one for each label, a single conditional probability corresponding to the positive class $+1$ is used. That is, let $\eta(x) = \mathcal{D}(Y = +1 \mid X = x)$ denote the conditional probability of $Y = +1$ given $X = x$. The conditional error can then be expressed as:
$$\mathcal{C}_\ell(h,x) = \mathbb{E}_y[\ell(h,x,y)\mid x] = \eta(x)\ell(h,x,+1) + (1-\eta(x))\ell(h,x,-1).$$

For convenience, we also define, for any $p \in [0,1]$, $\mathcal{C}_\ell(h,x,p) = p\ell(h,x,+1) + (1-p)\ell(h,x,-1)$, $\mathcal{C}_{\ell,\mathcal{H}}^*(x,p) = \inf_{h\in\mathcal{H}} \mathcal{C}_\ell(h,x,p)$ and $\Delta\mathcal{C}_{\ell,\mathcal{H}}(h,x,p) = \mathcal{C}_\ell(h,x,p) - \inf_{h\in\mathcal{H}} \mathcal{C}_\ell(h,x,p)$. Thus, we have $\Delta\mathcal{C}_{\ell,\mathcal{H}}(h,x,\eta(x)) = \Delta\mathcal{C}_{\ell,\mathcal{H}}(h,x)$.

To simplify matters, we will use the same notation for binary and multi-class classification, such as $\mathcal{Y}$ for the label space or $\mathcal{H}$ for a hypothesis set. We rely on the reader to adapt to the appropriate definitions based on the context.

**Estimation, approximation, and excess errors.** For a hypothesis $h$, the difference $\mathcal{E}_\ell(h) - \mathcal{E}_\ell^*(\mathcal{H}_{\text{all}})$ is known as the *excess error*. It can be decomposed into the sum of two terms, the *estimation error*, $(\mathcal{E}_\ell(h) - \mathcal{E}_\ell^*(\mathcal{H}))$ and the *approximation error* $\mathcal{A}_\ell(\mathcal{H}) = (\mathcal{E}_\ell^*(\mathcal{H}) - \mathcal{E}_\ell^*(\mathcal{H}_{\text{all}}))$:
$$\mathcal{E}_\ell(h) - \mathcal{E}_\ell^*(\mathcal{H}_{\text{all}}) = (\mathcal{E}_\ell(h) - \mathcal{E}_\ell^*(\mathcal{H})) + (\mathcal{E}_\ell^*(\mathcal{H}) - \mathcal{E}_\ell^*(\mathcal{H}_{\text{all}})). \tag{1}$$

A fundamental result for a pointwise loss function $\ell$ is that the Bayes error and the approximation error admit the following simpler expressions. We give a concise proof of this lemma in Appendix B, where we establish the measurability of the function $x \mapsto \mathcal{C}_\ell^*(\mathcal{H}_{\text{all}},x)$.

**Lemma 2.1.** *Let $\ell$ be a pointwise loss function. Then, the Bayes error and the approximation error can be expressed as follows: $\mathcal{E}_\ell^*(\mathcal{H}_{\text{all}}) = \mathbb{E}_x[\mathcal{C}_\ell^*(\mathcal{H}_{\text{all}},x)]$ and $\mathcal{A}_\ell(\mathcal{H}) = \mathcal{E}_\ell^*(\mathcal{H}) - \mathbb{E}_x[\mathcal{C}_\ell^*(\mathcal{H}_{\text{all}},x)]$.*

For restricted hypothesis sets ($\mathcal{H} \neq \mathcal{H}_{\text{all}}$), the infimum's super-additivity implies that $\mathcal{E}_\ell^*(\mathcal{H}) \geq \mathbb{E}_x[\mathcal{C}_\ell^*(\mathcal{H},x)]$. This inequality is generally strict, and the difference, $\mathcal{E}_\ell^*(\mathcal{H}) - \mathbb{E}_x[\mathcal{C}_\ell^*(\mathcal{H},x)]$, plays a crucial role in our analysis.

## 3 $\mathcal{H}$-consistency bounds

A widely used notion of consistency is that of *Bayes-consistency* given below [Steinwart, 2007].

**Definition 3.1 (Bayes-consistency).** A loss function $\ell_1$ is *Bayes-consistent* with respect to a loss function $\ell_2$, if for any distribution $\mathcal{D}$ and any sequence $\{h_n\}_{n\in\mathbb{N}} \subset \mathcal{H}_{\text{all}}$, $\lim_{n\to+\infty} \mathcal{E}_{\ell_1}(h_n) - \mathcal{E}_{\ell_1}^*(\mathcal{H}_{\text{all}}) = 0$ implies $\lim_{n\to+\infty} \mathcal{E}_{\ell_2}(h_n) - \mathcal{E}_{\ell_2}^*(\mathcal{H}_{\text{all}}) = 0$.

Thus, when this property holds, asymptotically, a nearly optimal minimizer of $\ell_1$ over the family of all measurable functions is also a nearly optimal optimizer of $\ell_2$. But, Bayes-consistency does not supply any information about a hypothesis set $\mathcal{H}$ not containing the full family $\mathcal{H}_{\text{all}}$, that is a typical hypothesis set used for learning. Furthermore, it is only an asymptotic property and provides no convergence guarantee. In particular, it does not give any guarantee for approximate minimizers. Instead, we will consider upper bounds on the target estimation error expressed in terms of the surrogate estimation error, $\mathcal{H}$-*consistency bounds* [Awasthi et al., 2022b,a, Mao et al., 2023f], which account for the hypothesis set $\mathcal{H}$ adopted.

**Definition 3.2** ($\mathcal{H}$-**consistency bounds**). Given a hypothesis set $\mathcal{H}$, an $\mathcal{H}$-*consistency bound* relating the loss function $\ell_1$ to the loss function $\ell_2$ for a hypothesis set $\mathcal{H}$ is an inequality of the form

$$\forall h \in \mathcal{H}, \quad \mathcal{E}_{\ell_2}(h) - \mathcal{E}_{\ell_2}^*(\mathcal{H}) + \mathcal{M}_{\ell_2}(\mathcal{H}) \leq \Gamma\big(\mathcal{E}_{\ell_1}(h) - \mathcal{E}_{\ell_1}^*(\mathcal{H}) + \mathcal{M}_{\ell_1}(\mathcal{H})\big), \quad (2)$$

that holds for any distribution $\mathcal{D}$, where $\Gamma \colon \mathbb{R}_+ \to \mathbb{R}_+$ is a non-decreasing concave function with $\Gamma \geq 0$ [Awasthi et al., 2022b,a]. Here, $\mathcal{M}_{\ell_1}(\mathcal{H})$ and $\mathcal{M}_{\ell_2}(\mathcal{H})$ are *minimizability gaps* for the respective loss functions. The minimizability gap for a hypothesis set $\mathcal{H}$ and loss function $\ell$ is denoted by $\mathcal{M}_\ell(\mathcal{H})$ and defined as: $\mathcal{M}_\ell(\mathcal{H}) = \mathcal{E}_\ell^*(\mathcal{H}) - \mathbb{E}_x[\mathcal{C}_\ell^*(\mathcal{H}, x)]$. It quantifies the discrepancy between the best possible expected loss within a hypothesis class and the expected infimum of pointwise expected losses. This gap is always non-negative: $\mathcal{M}_\ell(\mathcal{H}) = \inf_{h \in \mathcal{H}} \mathbb{E}_x[\mathcal{C}_\ell(h, x)] - \mathbb{E}_x[\inf_{h \in \mathcal{H}} \mathcal{C}_\ell(\mathcal{H}, x)] \geq 0$, by the infimum's super-additivity, and is bounded above by the approximation error $\mathcal{A}_\ell(\mathcal{H}) = \inf_{h \in \mathcal{H}} \mathbb{E}_x[\mathcal{C}_\ell(h, x)] - \mathbb{E}_x[\inf_{h \in \mathcal{H}_{\text{all}}} \mathcal{C}_\ell(\mathcal{H}, x)]$. We further study the key role of minimizability gaps in $\mathcal{H}$-consistency bounds and their properties in Section 6 and Appendix D. As shown in Appendix C, under general assumptions, minimizability gaps are essential quantities required in any bound that relates the estimation errors of two loss functions with an arbitrary hypothesis set $\mathcal{H}$.

Thus, an $\mathcal{H}$-consistency bound provides the guarantee that when the surrogate estimation loss $\mathcal{E}_\ell(h) - \mathcal{E}_\ell^*(\mathcal{H})$ is minimized to $\epsilon$, the following upper bound holds for the zero-one estimation error:

$$\mathcal{E}_\ell(h) - \mathcal{E}_\ell^*(\mathcal{H}) \leq \Gamma(\epsilon + \mathcal{M}_\ell(\mathcal{H})) - \mathcal{M}_{\ell_{0-1}}(\mathcal{H}) \leq \Gamma(\epsilon) + \Gamma(\mathcal{M}_\ell(\mathcal{H})) - \mathcal{M}_{\ell_{0-1}}(\mathcal{H}),$$

where the second inequality follows from the sub-additivity of a concave function $\Gamma$ over $\mathbb{R}_+$. We will demonstrate that, for smooth surrogate losses, $\Gamma(\epsilon)$ scales as $\sqrt{\epsilon}$. Note, however, that, while $\Gamma(\epsilon)$ tends to zero when $\epsilon \to 0$ for functions $\Gamma$ derived in $\mathcal{H}$-consistency bounds, the remaining terms in the bound are constant. This is not surprising as, in general, minimizing the surrogate estimation error to zero *cannot* guarantee that the zero-one estimation error will also converge to zero. This is well-known, for example, in the case of linear models [Ben-David et al., 2012]. Instead, an $\mathcal{H}$-consistency bound provides the tightest possible upper bound on the estimation error for the zero-one loss when the surrogate estimation error is minimized.

The upper bound simplifies to $\Gamma(\epsilon)$ when the minimizability gaps are zero, which occurs when either $\mathcal{H} = \mathcal{H}_{\text{all}}$ (the set of all measurable functions) or in realizable cases, which are particularly relevant to the practical use of complex neural networks in applications. In Appendix I, we examine more general cases of small minimizability gaps, taking into account the complexity of $\mathcal{H}$ and the distribution.

Our results cover in particular the special case of excess bounds ($\mathcal{H} = \mathcal{H}_{\text{all}}$). Let us emphasize that, for $\mathcal{H} \neq \mathcal{H}_{\text{all}}$, $\mathcal{H}$-consistency bounds offer tighter and more favorable guarantees on the estimation error compared to those derived from excess bounds analysis alone (see Appendix F).

When $\ell_2 = \ell_{0-1}$, the zero-one loss, we say that $\mathcal{T}$ is the $\mathcal{H}$-*estimation error transformation function of a surrogate loss* $\ell$ if the following holds:

$$\forall h \in \mathcal{H}, \quad \mathcal{T}\big(\mathcal{E}_{\ell_{0-1}}(h) - \mathcal{E}_{\ell_{0-1}}^*(\mathcal{H}) + \mathcal{M}_{\ell_{0-1}}(\mathcal{H})\big) \leq \mathcal{E}_\ell(h) - \mathcal{E}_\ell^*(\mathcal{H}) + \mathcal{M}_\ell(\mathcal{H}),$$

and the bound is *tight*. That is, for any $t \in [0, 1]$, there exists a hypothesis $h \in \mathcal{H}$ and a distribution such that $\mathcal{E}_{\ell_{0-1}}(h) - \mathcal{E}_{\ell_{0-1}}^*(\mathcal{H}) + \mathcal{M}_{\ell_{0-1}}(\mathcal{H}) = t$ and $\mathcal{E}_\ell(h) - \mathcal{E}_\ell^*(\mathcal{H}) + \mathcal{M}_\ell(\mathcal{H}) = \mathcal{T}(t)$. An explicit form of $\mathcal{T}$ has been characterized for binary margin-based losses [Awasthi et al., 2022b], as well as comp-sum losses and constrained losses in multi-class classification [Mao et al., 2023b]. In the following sections, we will prove the property $\mathcal{T}(t) = \Theta(t^2)$ (under mild assumptions), demonstrating a square-root growth rate for $\mathcal{H}$-consistency bounds. Appendix E provides examples of $\mathcal{H}$-consistency bounds for both binary and multi-class classification. Our analysis also suggests choosing appropriately $\mathcal{H}$ and the function $\Gamma$ to ensure a small minimizability gap and to take into account the number of classes and other properties, as discussed in Section 6.

# 4 Binary classification

We consider the broad family of margin-based loss functions $\ell$ defined for any $h \in \mathcal{H}$, and $(x, y) \in \mathcal{X} \times \mathcal{Y}$ by $\ell(h, x, y) = \Phi(-yh(x))$, where $\Phi$ is a non-decreasing convex function upper-bounding the zero-one loss. Margin-based loss functions include most loss functions used in binary classification. As an example, $\Phi(u) = \log(1 + e^u)$ for the logistic loss or $\Phi(u) = \exp(u)$ for the exponential loss. We say that a hypothesis set $\mathcal{H}$ is *complete*, if for all $x \in \mathcal{X}$, we have $\{h(x): h \in \mathcal{H}\} = \mathbb{R}$. As shown by Awasthi et al. [2022b], the transformation $\mathcal{T}$ has the following form for complete hypothesis sets:

$$\mathcal{T}(t) := \inf_{u \leq 0} f_t(u) - \inf_{u \in \mathbb{R}} f_t(u).$$

Here, for any $t \in [0, 1]$, $f_t$ is defined by: $\forall u \in \mathbb{R}, f_t(u) = \frac{1-t}{2}\Phi(u) + \frac{1+t}{2}\Phi(-u)$. The following result is useful for proving the growth rate in binary classification.

**Theorem 4.1.** *Let $\mathcal{H}$ be a complete hypothesis set. Assume that $\Phi$ is convex and differentiable at zero and satisfies the inequality $\Phi'(0) > 0$. Then, the transformation $\mathcal{T}$ can be expressed as follows:*

$$\forall t \in [0, 1], \quad \mathcal{T}(t) = f_t(0) - \inf_{u \in \mathbb{R}} f_t(u).$$

*Proof.* By the convexity of $\Phi$, for any $t \in [0, 1]$ and $u \in \mathbb{R}_-$, we have

$$f_t(u) = \frac{1-t}{2}\Phi(u) + \frac{1+t}{2}\Phi(-u) \geq \Phi(0) - tu\Phi'(0) \geq \Phi(0).$$

Thus, we can write $\mathcal{T}(t) = \inf_{u \leq 0} f_t(u) - \inf_{u \in \mathbb{R}} f_t(u) \geq \Phi(0) - \inf_{u \in \mathbb{R}} f_t(u) = f_t(0) - \inf_{u \in \mathbb{R}} f_t(u)$, where equality is achieved when $u = 0$. $\qquad\square$

**Theorem 4.2** (Upper and lower bound for binary margin-based losses). *Let $\mathcal{H}$ be a complete hypothesis set. Assume that $\Phi$ is convex, twice continuously differentiable, and satisfies the inequalities $\Phi'(0) > 0$ and $\Phi''(0) > 0$. Then, the following property holds: $\mathcal{T}(t) = \Theta(t^2)$; that is, there exist positive constants $C > 0$, $c > 0$, and $T > 0$ such that $Ct^2 \geq \mathcal{T}(t) \geq ct^2$, for all $0 < t \leq T$.*

**Proof sketch** First, we demonstrate that, by applying the implicit function theorem, $\inf_{u \in \mathbb{R}} f_t(u)$ is attained uniquely by $a_t^*$, and that $a_t^*$ is continuously differentiable over $[0, \epsilon]$ for some $\epsilon > 0$. The minimizer $a_t^*$ satisfies the following condition: $f_t'(a_t^*) = \frac{1-t}{2}\Phi'(a_t^*) - \frac{1+t}{2}\Phi'(-a_t^*) = 0$. Specifically, at $t = 0$, we have $\Phi'(a_0^*) = \Phi'(-a_0^*)$. Then, by the convexity of $\Phi$ and monotonicity of the derivative $\Phi'$, we must have $a_0^* = 0$ and since $\Phi'$ is non-decreasing and $\Phi''(0) > 0$, we have $a_t^* > 0$ for all $t \in (0, \epsilon]$. Furthermore, since $a_t^*$ is a function of class $C^1$, we can differentiate this condition with respect to $t$ and take the limit $t \to 0$, which gives the following equality: $\frac{da_t^*}{dt}(0) = \frac{\Phi'(0)}{\Phi''(0)} > 0$. Since $\lim_{t \to 0} \frac{a_t^*}{t} = \frac{da_t^*}{dt}(0) = \frac{\Phi'(0)}{\Phi''(0)} > 0$, we have $a_t^* = \Theta(t)$. By Theorem 4.1 and Taylor's theorem with an integral remainder, $\mathcal{T}$ can be expressed as follows: for any $t \in [0, \epsilon]$, $\mathcal{T}(t) = f_t(0) - \inf_{u \in \mathbb{R}} f_t(u) = \int_0^{a_t^*} u f_t''(u)\, du = \int_0^{a_t^*} u \left[\frac{1-t}{2}\Phi''(u) + \frac{1+t}{2}\Phi''(-u)\right] du$. Since $\Phi''(0) > 0$ and $\Phi''$ is continuous, there is a non-empty interval $[-\alpha, +\alpha]$ over which $\Phi''$ is positive. Since $a_0^* = 0$ and $a_t^*$ is continuous, there exists a sub-interval $[0, \epsilon'] \subseteq [0, \epsilon]$ over which $a_t^* \leq \alpha$. Since $\Phi''$ is continuous, it admits a minimum and a maximum over any compact set and we can define $c = \min_{u \in [-\alpha, \alpha]} \Phi''(u)$ and $C = \max_{u \in [-\alpha, \alpha]} \Phi''(u)$. $c$ and $C$ are both positive since we have $\Phi''(0) > 0$. Thus, for $t$ in $[0, \epsilon']$, the following inequality holds: $C\frac{(a_t^*)^2}{2} = \int_0^{a_t^*} uC\, du \geq \mathcal{T}(t) = \int_0^{a_t^*} u\left[\frac{1-t}{2}\Phi''(u) + \frac{1+t}{2}\Phi''(-u)\right] du \geq \int_0^{a_t^*} uc\, du = c\frac{(a_t^*)^2}{2}$. This implies that $\mathcal{T}(t) = \Theta(t^2)$. The full proof is included in Appendix K.

Theorem 4.2 directly applies to excess error bounds as well, when $\mathcal{H} = \mathcal{H}_{\text{all}}$. Importantly, our lower bound requires weaker conditions than [Frongillo and Waggoner, 2021, Theorem 4], and our upper bound is entirely novel. This result demonstrates that the growth rate for these loss functions is precisely square-root, refining the "at least square-root" finding of these authors. It is known that polyhedral losses admit a linear grow rate [Frongillo and Waggoner, 2021]. Thus, a striking dichotomy emerges: $\mathcal{H}$-consistency bounds for polyhedral losses exhibit a linear growth rate, while they follow a square-root rate for smooth loss functions (see Appendix G for a detailed comparison).

# 5 Multi-class classification

In this section, we will study two families of surrogate losses in multi-class classification: comp-sum losses and constrained losses, defined in Section 5.1 and Section 5.2 respectively. Comp-sum losses and constrained losses are general and cover all loss functions commonly used in practice. We will consider any hypothesis set $\mathcal{H}$ that is *symmetric* and *complete*. We say that a hypothesis set is *symmetric* when it does not depend on a specific ordering of the classes, that is, when there exists a family $\mathcal{F}$ of functions $f$ mapping from $\mathcal{X}$ to $\mathbb{R}$ such that $\{[h(x,1),\ldots,h(x,n)] \colon h \in \mathcal{H}\} = \{[f_1(x),\ldots,f_n(x)] \colon f_1,\ldots,f_n \in \mathcal{F}\}$, for any $x \in \mathcal{X}$. We say that a hypothesis set $\mathcal{H}$ is *complete* if the set of scores it generates spans $\mathbb{R}$, that is, $\{h(x,y) \colon h \in \mathcal{H}\} = \mathbb{R}$, for any $(x,y) \in \mathcal{X} \times \mathcal{Y}$.

## 5.1 Comp-sum losses

Here, we consider comp-sum losses [Mao, Mohri, and Zhong, 2023f], defined as

$$\forall h \in \mathcal{H}, \forall (x,y) \times \mathcal{X} \times \mathcal{Y}, \quad \ell^{\mathrm{comp}}(h,x,y) = \Phi\left(\frac{e^{h(x,y)}}{\sum_{y' \in \mathcal{Y}} e^{h(x,y')}}\right),$$

where $\Phi\colon \mathbb{R} \to \mathbb{R}_+$ is a non-increasing function. For example, $\Phi$ can be chosen as the negative log function $u \mapsto -\log(u)$ for the comp-sum losses, which leads to the multinomial logistic loss. As shown by Mao, Mohri, and Zhong [2023b], for symmetric and complete hypothesis sets, the transformation $\mathcal{T}$ for the family of comp-sum losses can be characterized as follows.

**Theorem 5.1** (Mao et al. [2023b, Theorem 3]). *Let $\mathcal{H}$ be a symmetric and complete hypothesis set. Assume that $\Phi$ is convex, differentiable at $\frac{1}{2}$ and satisfies the inequality $\Phi'(\frac{1}{2}) < 0$. Then, the transformation $\mathcal{T}$ can be expressed as*

$$\mathcal{T}(t) = \inf_{\tau \in \left[\frac{1}{n},\frac{1}{2}\right]} \sup_{|u| \le \tau} \left\{ \Phi(\tau) - \frac{1-t}{2}\Phi(\tau+u) - \frac{1+t}{2}\Phi(\tau-u) \right\}.$$

Next, we will show that as with the binary case, for the comp-sum losses, the properties $\mathcal{T}(t) = \Omega(t^2)$ and $\mathcal{T}(t) = O(t^2)$ hold. We first introduce a generalization of the classical implicit function theorem where the function takes the value zero over a set of points parameterized by a compact set. We treat the special case of a function $F$ defined over $\mathbb{R}^3$ and denote by $(t,a,\tau) \in \mathbb{R}^3$ its arguments. The theorem holds more generally for the arguments being in $\mathbb{R}^{n_1} \times \mathbb{R}^{n_2} \times \mathbb{R}^{n_3}$ and with the condition on the partial derivative being non-zero replaced with a partial Jacobian being non-singular.

**Theorem 5.2** (Implicit function theorem with a compact set). *Let $F\colon \mathbb{R} \times \mathbb{R} \times \mathbb{R} \to \mathbb{R}$ be a continuously differentiable function in a neighborhood of $(0,0,\tau)$, for any $\tau$ in a non-empty compact set $\mathcal{C}$, with $F(0,0,\tau) = 0$. Then, if $\frac{\partial F}{\partial a}(0,0,\tau)$ is non-zero for all $\tau$ in $\mathcal{C}$, then, there exist a neighborhood $\mathcal{O}$ of $0$ and a unique function $\bar{a}$ defined over $\mathcal{O} \times \mathcal{C}$ that is continuously differentiable and satisfies*

$$\forall (t,\tau) \in \mathcal{O} \times \mathcal{C}, \quad F(t, \bar{a}(t,\tau), \tau) = 0.$$

*Proof.* By the implicit function theorem (see for example [Dontchev and Rockafellar, 2009]), for any $\tau \in \mathcal{C}$, there exists an open set $\mathcal{U}_\tau = (-t_\tau, +t_\tau) \times (\tau - \epsilon_\tau, \tau + \epsilon_\tau)$, ($t_\tau > 0$ and $\epsilon_\tau > 0$), and a unique function $\bar{a}_\tau \colon \mathcal{U}_\tau \to \mathbb{R}$ that is in $C^1$ and such that for all $(t,\tau) \in \mathcal{U}_\tau$, $F(t, \bar{a}_\tau(t), \tau) = 0$.

By the uniqueness of $\bar{a}_\tau$, for any $\tau \ne \tau'$ and $(t_1, \tau_1) \in \mathcal{U}_\tau \cap \mathcal{U}_{\tau'}$, we have $\bar{a}_\tau(t_1, \tau_1) = \bar{a}_{\tau'}(t_1, \tau_1)$. Thus, we can define a function $\bar{a}$ over $\mathcal{U} = \bigcup_{\tau \in \mathcal{C}} \mathcal{U}_\tau$ that is of class $C^1$ and such that for any $(t,\tau) \in \mathcal{U}$, $F(t, \bar{a}(t,\tau), \tau) = 0$.

Now, $\bigcup_{\tau \in \mathcal{C}} (\tau - \epsilon_\tau, \tau + \epsilon_\tau)$ is a cover of the compact set $\mathcal{C}$ via open sets. Thus, we can extract from it a finite cover $\bigcup_{\tau \in I} (\tau - \epsilon_\tau, \tau + \epsilon_\tau)$, for some finite cardinality set $I$. Define $(-t_0, +t_0) = \bigcap_{\tau \in I} (-t_\tau, +t_\tau)$, which is a non-empty open interval as an intersection of (embedded) open intervals containing zero. Then, $\bar{a}$ is continuously differentiable over $(-t_0, +t_0) \times \mathcal{C}$ and for any $(t,\tau) \in (-t_0, +t_0) \times \mathcal{C}$, we have $F(t, \bar{a}(t,\tau), \tau) = 0$. $\square$

**Theorem 5.3** (Upper and lower bound for comp-sum losses). *Assume that $\Phi$ is convex, twice continuously differentiable, and satisfies the properties $\Phi'(u) < 0$ and $\Phi''(u) > 0$ for any $u \in (0, \frac{1}{2}]$. Then, the following property holds: $\mathcal{T}(t) = \Theta(t^2)$.*

**Proof sketch** For any $\tau \in \left[\frac{1}{n}, \frac{1}{2}\right]$, define the function $\mathcal{T}_\tau$ by $\mathcal{T}_\tau(t) = f_{t,\tau}(0) - \inf_{|u| \leq \tau} f_{t,\tau}(u)$, where $f_{t,\tau}(u) = \frac{1-t}{2}\Phi_\tau(u) + \frac{1+t}{2}\Phi_\tau(-u)$, $t \in [0,1]$ and $\Phi_\tau(u) = \Phi(\tau + u)$.

We aim to establish a lower and upper bound for $\inf_{\tau \in [\frac{1}{n}, \frac{1}{2}]} \mathcal{T}_\tau(t)$. For any fixed $\tau \in \left[\frac{1}{n}, \frac{1}{2}\right]$, this situation is parallel to that of binary classification (Theorem 4.1 and Theorem 4.2), since we have $\Phi'_\tau(0) = \Phi'(\tau) < 0$ and $\Phi''_\tau(0) = \Phi''(\tau) > 0$. By Theorem 5.2 and the proof of Theorem 4.2, adopting a similar notation, while incorporating the $\tau$ subscript to distinguish different functions $\Phi_\tau$ and $f_{t,\tau}$, we can write $\forall t \in [0, t_0]$, $\mathcal{T}_\tau(t) = \int_0^{-a^*_{t,\tau}} u\left[\frac{1-t}{2}\Phi''_\tau(-u) + \frac{1+t}{2}\Phi''_\tau(u)\right] du$, where $a^*_{t,\tau}$ verifies $a^*_{0,\tau} = 0$ and $\frac{\partial a^*_{t,\tau}}{\partial t}(0) = \frac{\Phi'_\tau(0)}{\Phi''_\tau(0)} = c_\tau < 0$. Then, by further analyzing this equality, we can show the lower bound $\inf_{\tau \in [\frac{1}{n}, \frac{1}{2}]} -a^*_{t,\tau} = \Omega(t)$ and the upper bound $\sup_{\tau \in [\frac{1}{n}, \frac{1}{2}]} -a^*_{t,\tau} = O(t)$ for some $t \in [0, t_1]$, $t_1 > 0$. Finally, using the fact that $\Phi''$ reaches its maximum and minimum over a compact set, we obtain that $\mathcal{T}(t) = \inf_{\tau \in [\frac{1}{n}, \frac{1}{2}]} \mathcal{T}_\tau(t) = \Theta(t^2)$. The full proof is included in Appendix L.

Theorem 5.3 significantly extends Theorem 4.2 to multi-class comp-sum losses, which include the logistic loss or cross-entropy used with a softmax activation function. It shows that the growth rate of $\mathcal{H}$-consistency bounds for comp-sum losses is exactly square-root, provided that the auxiliary function $\Phi$ they are based upon is convex, twice continuously differentiable, and satisfies $\Phi'(u) < 0$ and $\Phi''(u) > 0$ for any $u \in \left(0, \frac{1}{2}\right]$, which holds for most loss functions used in practice.

## 5.2 Constrained losses

Here, we consider constrained losses (see [Lee et al., 2004]), defined as

$$\forall h \in \mathcal{H}, \forall (x, y) \times \mathcal{X} \times \mathcal{Y}, \quad \ell^{\text{cstnd}}(h, x, y) = \sum_{y' \neq y} \Phi(h(x, y')) \text{ subject to } \sum_{y \in \mathcal{Y}} h(x, y) = 0,$$

where $\Phi : \mathbb{R} \to \mathbb{R}_+$ is a non-decreasing function. On possible choice for $\Phi$ is the exponential function. As shown by Mao et al. [2023b], for symmetric and complete hypothesis sets, the transformation $\mathcal{T}$ for the family of constrained losses can be characterized as follows.

**Theorem 5.4** (Mao et al. [2023b, Theorem 11]). *Let $\mathcal{H}$ be a symmetric and complete hypothesis set. Assume that $\Phi$ is convex, differentiable at zero and satisfies the inequality $\Phi'(0) > 0$. Then, the transformation $\mathcal{T}$ can be expressed as*

$$\mathcal{T}(t) = \inf_{\tau \geq 0} \sup_{u \in \mathbb{R}} \left\{ \left(2 - \frac{1}{n-1}\right)\Phi(\tau) - \frac{2 - \frac{1}{n-1} - t}{2}\Phi(\tau + u) - \frac{2 - \frac{1}{n-1} + t}{2}\Phi(\tau - u) \right\}.$$

Next, we will show that for the constrained losses, the properties $\mathcal{T}(t) = \Omega(t)$ and $\mathcal{T}(t) = O(t)$ hold as well. Note that by Theorem 5.4, we have

$$\mathcal{T}\left(\left(2 - \frac{1}{n-1}\right)t\right) = \left(2 - \frac{1}{n-1}\right)\inf_{\tau \geq 0} \sup_{u \in \mathbb{R}} \left\{ \Phi(\tau) - \frac{1-t}{2}\Phi(\tau + u) - \frac{1+t}{2}\Phi(\tau - u) \right\}.$$

Therefore, to prove $\mathcal{T}(t) = \Theta(t^2)$, we only need to show

$$\inf_{\tau \geq 0} \sup_{u \in \mathbb{R}} \left\{ \Phi(\tau) - \frac{1-t}{2}\Phi(\tau + u) - \frac{1+t}{2}\Phi(\tau - u) \right\} = \Theta(t^2).$$

For simplicity, we assume that the infimum over $\tau \geq 0$ can be reached within some finite interval $[0, A]$, $A > 0$. This assumption holds for common choices of $\Phi$, as discussed in [Mao et al., 2023b]. Furthermore, as demonstrated in Appendix N, under certain conditions on $\Phi''$, the infimum over $\tau \in [0, A]$ is reached at zero for sufficiently small values of $t$. For specific examples, see [Mao et al., 2023b, Appendix D.3], where $\Phi(t) = e^t$ is considered.

**Theorem 5.5** (Upper and lower bound for constrained losses). *Assume that $\Phi$ is convex, twice continuously differentiable, and satisfies the properties $\Phi'(u) > 0$ and $\Phi''(u) > 0$ for any $u \geq 0$. Then, for any $A > 0$, the following property holds:*

$$\inf_{\tau \in [0, A]} \sup_{u \in \mathbb{R}} \left\{ \Phi(\tau) - \frac{1-t}{2}\Phi(\tau + u) - \frac{1+t}{2}\Phi(\tau - u) \right\} = \Theta(t^2).$$

**Proof sketch** For any $\tau \in [0, A]$, define the function $\mathcal{T}_\tau$ by $\mathcal{T}_\tau(t) = f_{t,\tau}(0) - \inf_{u \in \mathbb{R}} f_{t,\tau}(u)$, where $f_{t,\tau}(u) = \frac{1-t}{2}\Phi_\tau(u) + \frac{1+t}{2}\Phi_\tau(-u)$, $t \in [0,1]$ and $\Phi_\tau(u) = \Phi(\tau + u)$. We aim to establish a lower and upper bound for $\inf_{\tau \in [0,A]} \mathcal{T}_\tau(t)$. For any fixed $\tau \in [0, A]$, this situation is parallel to that of binary classification (Theorem 4.1 and Theorem 4.2), since we also have $\Phi'_\tau(0) = \Phi'(\tau) > 0$ and $\Phi''_\tau(0) = \Phi''(\tau) > 0$. By applying Theorem 5.2 and leveraging the proof of Theorem 4.2, adopting a similar notation, while incorporating the $\tau$ subscript to distinguish different functions $\Phi_\tau$ and $f_{t,\tau}$, we can write $\forall t \in [0, t_0]$, $\mathcal{T}_\tau(t) = \int_0^{a^*_{t,\tau}} u\left[\frac{1-t}{2}\Phi''_\tau(u) + \frac{1+t}{2}\Phi''_\tau(-u)\right] du$, where $a^*_{t,\tau}$ verifies $a^*_{0,\tau} = 0$ and $\frac{\partial a^*_{t,\tau}}{\partial t}(0) = \frac{\Phi'_\tau(0)}{\Phi''_\tau(0)} = c_\tau > 0$. Then, by further analyzing this equality, we can show the lower bound $\inf_{\tau \in [0,A]} a^*_{t,\tau} = \Omega(t)$ and the upper bound $\sup_{\tau \in [0,A]} a^*_{t,\tau} = O(t)$ for some $t \in [0, t_1]$, $t_1 > 0$. Finally, using the fact that $\Phi''$ reaches its maximum and minimum over some compact set, we obtain that $\mathcal{T}(t) = \inf_{\tau \in [0,A]} \mathcal{T}_\tau(t) = \Theta(t^2)$. The full proof is included in Appendix M.

Theorem 5.5 significantly expands our findings to multi-class constrained losses. It demonstrates that, under some assumptions, which are commonly satisfied by smooth constrained losses used in practice, constrained loss $\mathcal{H}$-consistency bounds also exhibit a square-root growth rate.

# 6 Minimizability gaps

As shown in Sections 4 and 5, $\mathcal{H}$-consistency bounds for smooth loss functions in both binary and multi-class classification all admit a square-root growth rate near zero. In this section, we start by examining how the number of classes impacts these bounds. We then turn our attention to the minimizability gaps, which are the only distinguishing factors between the bounds.

## 6.1 Dependency on number of classes

Even with identical growth rates, surrogate losses can vary in their $\mathcal{H}$-consistency bounds due to the number of classes. This factor becomes crucial to consider when the class count is large. Consider the family of comp-sum loss functions $\ell_\tau^{\mathrm{comp}}$ with $\tau \in [0, 2)$, defined as

$$\ell_\tau^{\mathrm{comp}}(h, x, y) = \Phi^\tau\left(\frac{e^{h(x,y)}}{\sum_{y' \in \mathcal{Y}} e^{h(x,y')}}\right) = \begin{cases} \frac{1}{1-\tau}\left(\left[\sum_{y' \in \mathcal{Y}} e^{h(x,y')-h(x,y)}\right]^{1-\tau} - 1\right) & \tau \neq 1, \tau \in [0, 2) \\ \log\left(\sum_{y' \in \mathcal{Y}} e^{h(x,y')-h(x,y)}\right) & \tau = 1, \end{cases}$$

where $\Phi^\tau(u) = -\log(u)1_{\tau=1} + \frac{1}{1-\tau}\left(u^{\tau-1} - 1\right)1_{\tau \neq 1}$, for any $\tau \in [0, 2)$. Mao et al. [2023f, Eq. (7) & Theorem 3.1], established the following bound for any $h \in \mathcal{H}$ and $\tau \in [1, 2)$,

$$\mathcal{R}_{\ell_{0-1}}(h) - \mathcal{R}^*_{\ell_{0-1}}(\mathcal{H}) \leq \widetilde{\Gamma}_\tau\left(\mathcal{R}_{\ell_\tau^{\mathrm{comp}}}(h) - \mathcal{R}^*_{\ell_\tau^{\mathrm{comp}}}(\mathcal{H}) + \mathcal{M}_{\ell_\tau^{\mathrm{comp}}}(\mathcal{H})\right) - \mathcal{M}_{\ell_{0-1}}(\mathcal{H}),$$

where $\widetilde{\Gamma}_\tau(t) = \sqrt{2n^{\tau-1}t}$. Thus, while all these loss functions show square-root growth, the number of classes acts as a critical scaling factor.

## 6.2 Comparison across comp-sum losses

In Appendix H, we compare minimizability gaps cross comp-sum losses. We will see that minimizability gaps decrease as $\tau$ increases. This might suggest favoring $\tau$ close to 2. But when accounting for $n$, $\ell_\tau^{\mathrm{comp}}$ with $\tau = 1$ (logistic loss) is optimal since $n$ then vanishes. Thus, both class count and minimizability gaps are essential in loss selection. In Appendix J, we will show that the minimizability gaps can become zero or relatively small under certain conditions in multi-class classification. In such scenarios, the logistic loss is favored, which can partly explain its widespread practical application.

## 6.3 Small surrogate minimizability gaps

While minimizability gaps vanish in special scenarios (e.g., unrestricted hypothesis sets, best-in-class error matching Bayes error), we now seek broader conditions for zero or small surrogate minimizability gaps to make our bounds more meaningful.

Due to space constraints, we focus on binary classification here, with multi-class results given in Appendix J. We address pointwise surrogate losses which take the form $\ell(h(x), y)$ for a labeled point

$(x, y)$. We write $A = \{h(x) : h \in \mathcal{H}\}$ to denote the set of predictor values at $x$, which we assume to be independent of $x$. All proofs for this section are presented in Appendix I.

**Deterministic scenario**. We first consider the deterministic scenario, where the conditional probability $p(y|x)$ is either zero or one. For a deterministic distribution, we denote by $\mathcal{X}_+$ the subset of $\mathcal{X}$ over which the label is +1 and by $\mathcal{X}_-$ the subset of $\mathcal{X}$ over which the label is −1. For convenience, let $\ell_+ = \inf_{\alpha \in A} \ell(\alpha, +1)$ and $\ell_- = \inf_{\alpha \in A} \ell(\alpha, -1)$.

**Theorem 6.1.** *Assume that $\mathcal{D}$ is deterministic and that the best-in-class error is achieved by some $h^* \in \mathcal{H}$. Then, the minimizability gap is null, $\mathcal{M}(\mathcal{H}) = 0$, iff*

$$\ell(h^*(x), +1) = \ell_+ \text{ a.s. over } \mathcal{X}_+, \quad \ell(h^*(x), -1) = \ell_- \text{ a.s. over } \mathcal{X}_-.$$

*If further $\alpha \mapsto \ell(\alpha, +1)$ and $\alpha \mapsto \ell(\alpha, -1)$ are injective and $\ell_+ = \ell(\alpha_+, +1)$, $\ell_- = \ell(\alpha_-, -1)$, then, the condition is equivalent to $h^*(x) = \alpha_+ 1_{x \in \mathcal{X}_+} + \alpha_- 1_{x \in \mathcal{X}_-}$ Furthermore, the minimizability gap is bounded by $\epsilon$ iff $p(\mathbb{E}[\ell(h^*(x), +1) \mid y = +1] - \ell_+) + (1 - p)(\mathbb{E}[\ell(h^*(x), -1) \mid y = -1] - \ell_-) \leq \epsilon$. In particular, the condition implies:*

$$\mathbb{E}[\ell(h^*(x), +1) \mid y = +1] - \ell_+ \leq \frac{\epsilon}{p} \quad and \quad \mathbb{E}[\ell(h^*(x), -1) \mid y = -1] - \ell_- \leq \frac{\epsilon}{1 - p}.$$

The theorem suggests that, under those assumptions, for the surrogate minimizability gap to be zero, the best-in-class hypothesis must be piecewise constant with specific values on $\mathcal{X}_+$ and $\mathcal{X}_-$. The existence of such a hypothesis in $\mathcal{H}$ depends both on the complexity of the decision surface separating $\mathcal{X}_+$ and $\mathcal{X}_-$ and on that of the hypothesis set $\mathcal{H}$. More generally, when the best-in-class classifier $\epsilon$-approximates $\alpha_+$ over $\mathcal{X}_+$ and $\alpha_-$ over $\mathcal{X}_-$, then the minimizability gap is bounded by $\epsilon$. As an example, when the decision surface is a hyperplane, a hypothesis set of linear functions combined with a sigmoid activation function can provide such a good approximation (see Figure 1 for an illustration in a simple case).

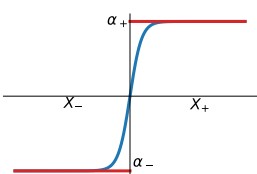

Figure 1: Approximation provided by sigmoid activation function.

**Stochastic scenario**. Here, we present a general result that is a direct extension of that of the deterministic scenario. We show that the minimizability gap is zero when there exists $h^* \in \mathcal{H}$ that matches $\alpha^*(x)$ for all $x$, where $\alpha^*(x)$ is the minimizer of the conditional error. We also show that the minimizability gap is bounded by $\epsilon$ when there exists $h^* \in \mathcal{H}$ whose conditional error $\epsilon$-approximates best-in-class conditional error for all $x$.

**Theorem 6.2.** *The best-in-class error is achieved by some $h^* \in \mathcal{H}$ and the minimizability gap is null, $\mathcal{M}(\mathcal{H}) = 0$, iff there exists $h^* \in \mathcal{H}$ such that for all $x$,*

$$\mathbb{E}_y[\ell(h^*(x), y) \mid x] = \inf_{\alpha \in A} \mathbb{E}_y[\ell(\alpha, y) \mid x] \text{ a.s. over } \mathcal{X}. \tag{3}$$

*If further $\alpha \mapsto \mathbb{E}_y[\ell(\alpha, y) \mid x]$ is injective and $\inf_{\alpha \in A} \mathbb{E}_y[\ell(\alpha, y) \mid x] = \mathbb{E}_y[\ell(\alpha^*(x), y) \mid x]$, then, the condition is equivalent to $h^*(x) = \alpha^*(x)$ a.s. for $x \in \mathcal{X}$. Furthermore, the minimizability gap is bounded by $\epsilon$, $\mathcal{M}(\mathcal{H}) \leq \epsilon$, iff there exists $h^* \in \mathcal{H}$ such that*

$$\mathbb{E}_x\left[ \mathbb{E}_y[\ell(h^*(x), y) \mid x] - \inf_{\alpha \in A} \mathbb{E}_y[\ell(\alpha, y) \mid x] \right] \leq \epsilon. \tag{4}$$

In deterministic settings, condition (4) coincides with that of Theorem 6.1. However, in stochastic scenarios, the existence of such a hypothesis depends on both decision surface complexity and the conditional distribution's properties. For illustration, see Appendix J.3 where we analyze the exponential, logistic (binary), and multi-class logistic losses.

We thoroughly analyzed minimizability gaps, comparing them across comp-sum losses, and identifying conditions for zero or small gaps, which help inform surrogate loss selection. In Appendix F, we show the crucial role of minimizability gaps in comparing excess bounds with $\mathcal{H}$-consistency bounds. Importantly, combining $\mathcal{H}$-consistency bounds with surrogate loss Rademacher complexity bounds yields zero-one loss (estimation) learning bounds for surrogate loss minimizers (see Appendix O).

## 7 Conclusion

We established a universal square-root growth rate for the widely-used class of smooth surrogate losses in both binary and multi-class classification. This underscores the minimizability gap as a crucial discriminator among surrogate losses. Our detailed analysis of these gaps can provide guidance for loss selection.

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

# Contents of Appendix

# A  Related work

The Bayes-consistency of surrogate losses has been extensively studied in the context of binary classification. Zhang [2004a], Bartlett et al. [2006] and Steinwart [2007] established Bayes-consistency for various convex loss functions, including margin-based surrogates. They also introduced excess error bounds (or surrogate regret bounds) for margin-based surrogates. Other works include Chen et al. [2004], which specifically studied the SVM $q$-norm soft margin loss and established a square-root excess error bound with an optimal constant for that specific family, and Reid and Williamson [2009], which established tight excess error bounds for proper losses in binary classification.

The Bayes-consistency of several surrogate loss function families in the context of multi-class classification has also been studied by Zhang [2004b] and Tewari and Bartlett [2007]. Zhang [2004b] established a series of results for various multi-class classification formulations, including negative results for multi-class hinge loss functions [Crammer and Singer, 2001], as well as positive results for the sum exponential loss [Weston and Watkins, 1999, Awasthi, Mao, Mohri, and Zhong, 2022a], the (multinomial) logistic loss [Verhulst, 1838, 1845, Berkson, 1944, 1951], and the constrained losses [Lee et al., 2004]. Later, Tewari and Bartlett [2007] adopted a different geometric method to analyze Bayes-consistency, yielding similar results for these loss function families. Steinwart [2007] developed general tools to characterize Bayes consistency for both binary and multi-class classification. Additionally, excess error bounds have been derived by Pires et al. [2013] for a family of constrained losses and by Duchi et al. [2018] for loss functions related to generalized entropies.

For a surrogate loss $\ell$, an excess error bound holds for any predictor $h$ and has the form $\mathcal{E}_{\ell_{0-1}}(h) - \mathcal{E}^*_{\ell_{0-1}} \leq \Psi(\mathcal{E}_\ell(h) - \mathcal{E}^*_\ell)$, where $\mathcal{E}_{\ell_{0-1}}(h)$ and $\mathcal{E}_\ell(h)$ represent the expected losses of $h$ for the zero-one loss and surrogate loss respectively, $\mathcal{E}^*_{\ell_{0-1}}$ and $\mathcal{E}^*_\ell$ the Bayes errors for the zero-one and surrogate loss respectively, and $\Psi$ a non-decreasing function.

The *growth rate* of excess error bounds, that is the behavior of function $\Psi$ near zero, has gained attention in recent research [Mahdavi et al., 2014, Zhang et al., 2021, Frongillo and Waggoner, 2021, Bao, 2023]. Mahdavi et al. [2014] examined the growth rate for *smoothed hinge losses* in binary classification, demonstrating that smoother losses result in worse growth rates. The optimal rate is achieved with the standard hinge loss, which exhibits linear growth. Zhang et al. [2021] tied the growth rate of excess error bounds in binary classification to two properties of the surrogate loss function: consistency intensity and conductivity. These metrics enable comparisons of growth rates across different surrogates. But, can we establish lower and upper bounds for the growth rate of excess error bounds under specific regularity conditions?

Frongillo and Waggoner [2021] pioneered research on this question in binary classification settings. They established a critical square-root lower bound for excess error bounds when a surrogate loss is locally strongly convex and has a locally Lipschitz gradient. Additionally, they demonstrated a linear excess error bound for Bayes-consistent polyhedral loss functions (convex and piecewise-linear) [Finocchiaro et al., 2019] (see also [Lapin et al., 2016, Ramaswamy et al., 2018, Yu and Blaschko, 2018, Yang and Koyejo, 2020]). More recently, Bao [2023] complemented these results by showing that proper losses associated with Shannon entropy, exponential entropy, spherical entropy, squared $\alpha$-norm entropies and $\alpha$-polynomial entropies, with $\alpha > 1$, also exhibit a square-root lower bound for excess error bounds relative to the $\ell_1$-distance.

However, while Bayes-consistency and excess error bounds are valuable, they are not sufficiently informative, as they are established for the family of all measurable functions and disregard the crucial role played by restricted hypothesis sets in learning. As pointed out by Long and Servedio [2013], in some cases, minimizing Bayes-consistent losses can result in constant expected error, while minimizing inconsistent losses can yield an expected loss approaching zero. To address this limitation, the authors introduced the concept of *realizable $\mathcal{H}$-consistency*, further explored by Kuznetsov et al. [2014] and Zhang and Agarwal [2020]. Nonetheless, these guarantees are only asymptotic and rely on a strong realizability assumption that typically does not hold in practice.

Recent research by Awasthi, Mao, Mohri, and Zhong [2022b,a] and Mao, Mohri, and Zhong [2023f,c,e,b] has instead introduced and analyzed $\mathcal{H}$-*consistency bounds*. These bounds are more informative than Bayes-consistency since they are hypothesis set-specific and non-asymptotic. Their work covers broad families of surrogate losses in binary classication, multi-class classification, structured prediction, and abstention [Mao, Mohri, Mohri, and Zhong, 2023a]. Crucially, they provide upper bounds on the *estimation error* of the target loss, for example, the zero-one loss in classification,

that hold for any predictor $h$ within a hypothesis set $\mathcal{H}$. These bounds relate this estimation error to the surrogate loss estimation error. Their general form is: $\mathcal{E}_{\ell_{0-1}}(h) - \mathcal{E}^*_{\ell_{0-1}}(\mathcal{H}) \le f(\mathcal{E}_\ell(h) - \mathcal{E}^*_\ell(\mathcal{H}))$, where $\mathcal{E}^*_{\ell_{0-1}}(\mathcal{H})$ and $\mathcal{E}^*_\ell(\mathcal{H})$ represent the best-in-class expected losses for the zero-one and surrogate loss respectively, and $f$ is a non-decreasing function continuous at zero. $\mathcal{H}$-consistency bounds imply in particular excess error bounds, when the hypothesis set is taken to be the family of all measurable functions.

The authors have further analyzed $\mathcal{H}$-consistency bounds in structured prediction, ranking, and abstention.

Mao, Mohri, and Zhong [2023e] revealed limitations of existing structured prediction loss functions, demonstrating they lack Bayes-consistency. They introduced new surrogate loss families proven to benefit form $\mathcal{H}$-consistency bounds, thus also establishing Bayes-consistency. This complements earlier negative finding about the Bayes-consistency of Struct-SVM and positive results for quadratic surrogate (QS) losses or some non-smooth polyhedral-type loss functions [Osokin et al., 2017, Ciliberto et al., 2016, Blondel, 2019, Nowak-Vila et al., 2019, Nowak et al., 2019, 2020, Ciliberto et al., 2020, Nowak et al., 2022].

Mao et al. [2023c] showed that there are no meaningful $\mathcal{H}$-consistency bounds for general pairwise ranking and bipartite ranking surrogate losses with equicontinuous hypothesis sets, including linear models and neural networks. They proposed ranking with abstention as a solution. These results demonstrated that although these surrogate loss functions have been shown to be Bayes-consistent in various studies [Kotlowski et al., 2011, Menon and Williamson, 2014, Agarwal, 2014, Gao and Zhou, 2015, Uematsu and Lee, 2017], they are, in fact, not $\mathcal{H}$-consistent.

Mao et al. [2023a] applied $\mathcal{H}$-consistency bounds to two-stage learning to defer scenarios, designing new surrogate losses. This complemented the Bayes-consistent surrogate losses in the single-stage scenario of learning to defer [Mozannar and Sontag, 2020, Verma and Nalisnick, 2022, Verma et al., 2023, Mozannar et al., 2023] or learning with abstention [Bartlett and Wegkamp, 2008, Yuan and Wegkamp, 2010, Cortes et al., 2016a,b, Ramaswamy et al., 2018, Ni et al., 2019, Charoenphakdee et al., 2021, Cao et al., 2022, Cortes et al., 2023].

Mao, Mohri, and Zhong [2024d] recently established enhanced $\mathcal{H}$-consistency bounds based on more general inequalities relating conditional regrets. They derived more favorable distribution- and predictor-dependent bounds in various scenarios including standard multi-class classification, binary and multi-class classification under Tsybakov noise conditions, and bipartite ranking. $\mathcal{H}$-consistency bounds have also been studied in the scenario of adversarial robustness [Awasthi et al., 2021a,b, 2023a,b], bounded regression [Mao et al., 2024e], regression with multi-expert deferral [Mao et al., 2024h], top-$k$ classification [Cortes et al., 2024], multi-label learning [Mao et al., 2024f], score-based abstention [Mao et al., 2024c], predictor-rejector abstention [Mao et al., 2024b], learning to abstain with a fixed predictor with application in decontextualization [Mohri et al., 2024], ranking with abstention [Mao et al., 2023d], realizable learning to defer [Mao et al., 2024g], and learning to defer with multiple experts [Mao et al., 2024a].

This papers presents a characterization of the growth rate of $\mathcal{H}$-consistency bounds, that is how quickly the functions $f$ increase near zero, both in binary and multi-class classification.

# B  Pointwise loss functions - Proof of Lemma 2.1

**Lemma 2.1.** *Let $\ell$ be a pointwise loss function. Then, the Bayes error and the approximation error can be expressed as follows:* $\mathcal{E}^*_\ell(\mathcal{H}_{\mathrm{all}}) = \mathbb{E}_x[\mathcal{C}^*_\ell(\mathcal{H}_{\mathrm{all}}, x)]$ *and* $\mathcal{A}_\ell(\mathcal{H}) = \mathcal{E}^*_\ell(\mathcal{H}) - \mathbb{E}_x[\mathcal{C}^*_\ell(\mathcal{H}_{\mathrm{all}}, x)]$.

*Proof.* By definition, for a pointwise loss function $\ell$, there exists a measurable function $\hat{\ell}:\mathbb{R}^n \times \mathcal{Y} \to \mathbb{R}_+$ such that $\ell(h, x, y) = \hat{\ell}(h(x), y)$, where $h(x) = [h(x, 1), \ldots, h(x, n)]$ is the score vector of the predictor $h$. Thus, the following inequality holds:

$$\mathcal{C}^*_\ell(\mathcal{H}_{\mathrm{all}}, x) = \inf_{h \in \mathcal{H}} \mathbb{E}_y[\ell(h, x, y) \mid x] = \inf_{\alpha \in \mathbb{R}^n} \mathbb{E}_y[\hat{\ell}(\alpha, y) \mid x].$$

Since $\hat{\ell}:(\alpha, y) \mapsto \mathbb{R}_+$ is measurable, the function $(\alpha, x) \mapsto \mathbb{E}_y[\hat{\ell}(\alpha, y) \mid x]$ is also measurable. We now show that the function $x \mapsto \mathcal{C}^*_\ell(\mathcal{H}_{\mathrm{all}}, x) = \inf_{\alpha \in \mathbb{R}} \mathbb{E}_y[\hat{\ell}(\alpha, y) \mid x]$ is also measurable.

To do this, we consider for any $\beta > 0$, the set $\left\{ x : \inf_{\alpha \in \mathbb{R}} \mathbb{E}_y \left[ \hat{\ell}(\alpha, y) \mid x \right] < \beta \right\}$ which can be expressed as

$$\left\{ x : \inf_{\alpha \in \mathbb{R}^n} \mathbb{E}_y \left[ \hat{\ell}(\alpha, y) \mid x \right] < \beta \right\} = \left\{ x : \exists \alpha \in \mathbb{R}^n \text{ such that } \mathbb{E}_y \left[ \hat{\ell}(\alpha, y) \mid x \right] < \beta \right\}$$

$$= \Pi_{\mathcal{X}} \left\{ (\alpha, x) : \mathbb{E}_y \left[ \hat{\ell}(\alpha, y) \mid x \right] < \beta \right\},$$

where $\Pi_{\mathcal{X}}$ is the projection onto $\mathcal{X}$. By the measurable projection theorem, $x \mapsto \inf_{\alpha \in \mathbb{R}^n} \mathbb{E}_y \left[ \hat{\ell}(\alpha, y) \mid x \right]$ is measurable. Then, since a pointwise difference of measurable functions is measurable, for all $n \in \mathbb{N}$, the set $\left\{ (\alpha, x) : \mathbb{E}_y \left[ \hat{\ell}(\alpha, y) \mid x \right] < \inf_{\alpha \in \mathbb{R}^n} \mathbb{E}_y \left[ \hat{\ell}(\alpha, y) \mid x \right] + \frac{1}{n} \right\}$ is measurable. Thus, by Kuratowski and Ryll-Nardzewski [1965]'s measurable selection theorem, for all $n \in \mathbb{N}$, there exists a measurable function $h_n : x \mapsto \alpha \in \mathbb{R}^n$ such that the following holds:

$$\mathcal{C}_\ell(h_n, x) = \mathbb{E}_y \left[ \hat{\ell}(\alpha, y) \mid x \right] < \inf_{\alpha \in \mathbb{R}^n} \mathbb{E}_y \left[ \hat{\ell}(\alpha, y) \mid x \right] + \frac{1}{n} = \mathcal{C}_\ell^*(\mathcal{H}_{\mathrm{all}}, x) + \frac{1}{n}.$$

Therefore, we have

$$\mathcal{E}_\ell^*(\mathcal{H}_{\mathrm{all}}) \leq \mathbb{E}_x \left[ \mathcal{C}_\ell(h_n, x) \right] \leq \mathbb{E}_x \left[ \mathcal{C}_\ell^*(\mathcal{H}_{\mathrm{all}}, x) \right] + \frac{1}{n} \leq \mathcal{E}_\ell^*(\mathcal{H}_{\mathrm{all}}) + \frac{1}{n}.$$

By taking the limit $n \to +\infty$, we obtain $\mathcal{E}_\ell^*(\mathcal{H}_{\mathrm{all}}) = \mathbb{E}_x \left[ \mathcal{C}_\ell^*(\mathcal{H}_{\mathrm{all}}, x) \right]$. By definition, $\mathcal{A}_\ell(\mathcal{H}) = \mathcal{E}_\ell^*(\mathcal{H}) - \mathcal{E}_\ell^*(\mathcal{H}_{\mathrm{all}}) = \mathcal{E}_\ell^*(\mathcal{H}) - \mathbb{E}_x \left[ \mathcal{C}_\ell^*(\mathcal{H}_{\mathrm{all}}, x) \right]$. This completes the proof. $\qquad \square$

## C  General form of $\mathcal{H}$-consistency bounds

Fix a target loss function $\ell_2$ and a surrogate loss $\ell_1$. Given a hypothesis set $\mathcal{H}$, a bound relating the estimation errors of these loss functions admits the following form:

$$\forall h \in \mathcal{H}, \quad \mathcal{E}_{\ell_2}(h) - \mathcal{E}_{\ell_2}^*(\mathcal{H}) \leq \Gamma_{\mathcal{D}} \left( \mathcal{E}_{\ell_1}(h) - \mathcal{E}_{\ell_1}^*(\mathcal{H}) \right), \tag{5}$$

where, for any distribution $\mathcal{D}$, $\Gamma_{\mathcal{D}} : \mathbb{R}_+ \to \mathbb{R}_+$ is a non-decreasing function on $\mathbb{R}_+$. We will assume that $\Gamma_{\mathcal{D}}$ is concave. In particular, the bound should hold for any point mass distribution $\delta_x$, $x \in \mathcal{X}$. We will operate under the assumption that the same bound holds uniformly over $\mathcal{X}$ and thus that there exists a fixed concave function $\Gamma$ such that $\Gamma_{\delta_x} = \Gamma$ for all $x$.

Observe that for any point mass distribution $\delta_x$, the conditional loss and the expected loss coincide and therefore that we have $\mathcal{E}_{\ell_2}(h) - \mathcal{E}_{\ell_2}^*(\mathcal{H}) = \Delta \mathcal{C}_{\ell_2, \mathcal{H}}(\mathcal{H}, x)$, and similarly with $\ell_1$. Thus, we can write:

$$\forall h \in \mathcal{H}, \forall x \in \mathcal{X}, \quad \Delta \mathcal{C}_{\ell_2, \mathcal{H}}(h, x) \leq \Gamma(\Delta \mathcal{C}_{\ell_1, \mathcal{H}}(h, x)).$$

Therefore, by Jensen's inequality, for any distribution $\mathcal{D}$, we have

$$\forall h \in \mathcal{H}, \forall x \in \mathcal{X}, \quad \mathbb{E}_x \left[ \Delta \mathcal{C}_{\ell_2, \mathcal{H}}(h, x) \right] \leq \mathbb{E}_x \left[ \Gamma(\Delta \mathcal{C}_{\ell_1, \mathcal{H}}(h, x)) \right] \leq \Gamma \left( \mathbb{E}_x \left[ \Delta \mathcal{C}_{\ell_1, \mathcal{H}}(h, x) \right] \right).$$

Since $\mathbb{E}_x \left[ \Delta \mathcal{C}_{\ell_2, \mathcal{H}}(h, x) \right] = \mathcal{E}_{\ell_2}(h) - \mathcal{E}_{\ell_2}^*(\mathcal{H}) + \mathcal{M}_{\ell_2}(\mathcal{H})$ and similarly with $\ell_1$, we obtain the following bound for all distributions $\mathcal{D}$:

$$\forall h \in \mathcal{H}, \quad \mathcal{E}_{\ell_2}(h) - \mathcal{E}_{\ell_2}^*(\mathcal{H}) + \mathcal{M}_{\ell_2}(\mathcal{H}) \leq \Gamma \left( \mathcal{E}_{\ell_1}(h) - \mathcal{E}_{\ell_1}^*(\mathcal{H}) + \mathcal{M}_{\ell_1}(\mathcal{H}) \right). \tag{6}$$

This leads to the general form of $\mathcal{H}$-consistency bounds that we will be considering, which includes the key role of the minimizability gaps.

## D  Properties of minimizability gaps

By Lemma 2.1, for a pointwise loss function, we have $\mathcal{E}_\ell^*(\mathcal{H}_{\mathrm{all}}) = \mathbb{E}_x \left[ \mathcal{C}_\ell^*(\mathcal{H}_{\mathrm{all}}, x) \right]$, thus the minimizability gap vanishes for the family of all measurable functions.

**Lemma D.1.** *Let $\ell$ be a pointwise loss function. Then, we have $\mathcal{M}_\ell(\mathcal{H}_{\mathrm{all}}) = 0$.*

| $\Phi(u)$ | margin-based losses $\ell$ | $\mathcal{H}$-Consistency bounds |
|---|---|---|
| $e^u$ | $e^{-yh(x)}$ | $\mathcal{E}_{\ell_{0-1}}(h) - \mathcal{E}_{\ell_{0-1}}^*(\mathcal{H}) + \mathcal{M}_{\ell_{0-1}}(\mathcal{H}) \le \sqrt{2}(\mathcal{E}_\ell(h) - \mathcal{E}_\ell^*(\mathcal{H}) + \mathcal{M}_\ell(\mathcal{H}))^{\frac{1}{2}}$ |
| $\log(1 + e^u)$ | $\log(1 + e^{-yh(x)})$ | $\mathcal{E}_{\ell_{0-1}}(h) - \mathcal{E}_{\ell_{0-1}}^*(\mathcal{H}) + \mathcal{M}_{\ell_{0-1}}(\mathcal{H}) \le \sqrt{2}(\mathcal{E}_\ell(h) - \mathcal{E}_\ell^*(\mathcal{H}) + \mathcal{M}_\ell(\mathcal{H}))^{\frac{1}{2}}$ |
| $\max\{0, 1 + u\}^2$ | $\max\{0, 1 - yh(x)\}^2$ | $\mathcal{E}_{\ell_{0-1}}(h) - \mathcal{E}_{\ell_{0-1}}^*(\mathcal{H}) + \mathcal{M}_{\ell_{0-1}}(\mathcal{H}) \le (\mathcal{E}_\ell(h) - \mathcal{E}_\ell^*(\mathcal{H}) + \mathcal{M}_\ell(\mathcal{H}))^{\frac{1}{2}}$ |
| $\max\{0, 1 + u\}$ | $\max\{0, 1 - yh(x)\}$ | $\mathcal{E}_{\ell_{0-1}}(h) - \mathcal{E}_{\ell_{0-1}}^*(\mathcal{H}) + \mathcal{M}_{\ell_{0-1}}(\mathcal{H}) \le \mathcal{E}_\ell(h) - \mathcal{E}_\ell^*(\mathcal{H}) + \mathcal{M}_\ell(\mathcal{H})$ |

Table 1: Examples of $\mathcal{H}$-consistency bounds for binary margin-based losses.

Thus, in that case, (6) takes the following simpler form:

$$\forall h \in \mathcal{H}, \quad \mathcal{E}_{\ell_2}(h) - \mathcal{E}_{\ell_2}^*(\mathcal{H}_{\text{all}}) \le \Gamma\big(\mathcal{E}_{\ell_1}(h) - \mathcal{E}_{\ell_1}^*(\mathcal{H}_{\text{all}})\big). \tag{7}$$

In general, however, the minimizabiliy gap is non-zero for a restricted hypothesis set $\mathcal{H}$ and is therefore important to analyze. Let $\mathcal{I}_\ell(\mathcal{H})$ be the difference of pointwise infima $\mathcal{I}_\ell(\mathcal{H}) = \mathbb{E}_x[\mathcal{C}_\ell^*(\mathcal{H}, x) - \mathcal{C}_\ell^*(\mathcal{H}_{\text{all}}, x)]$, which is non-negative. Note that, for a pointwise loss function, the minimizability gap can be decomposed as follows in terms of the approximation error and the difference of pointwise infima:

$$\begin{aligned}
\mathcal{M}_\ell(\mathcal{H}) &= \mathcal{E}_\ell^*(\mathcal{H}) - \mathcal{E}_\ell^*(\mathcal{H}_{\text{all}}) + \mathcal{E}_\ell^*(\mathcal{H}_{\text{all}}) - \mathbb{E}_x\big[\mathcal{C}_\ell^*(\mathcal{H}, x)\big] \\
&= \mathcal{A}_\ell(\mathcal{H}) + \mathcal{E}_\ell^*(\mathcal{H}_{\text{all}}) - \mathbb{E}_x\big[\mathcal{C}_\ell^*(\mathcal{H}, x)\big] \\
&= \mathcal{A}_\ell(\mathcal{H}) - \mathcal{I}_\ell(\mathcal{H}) \le \mathcal{A}_\ell(\mathcal{H}).
\end{aligned}$$

Thus, the minimizabiliy gap can be upper bounded by the approximation error. It is however a finer quantity than the approximation error and can lead to more favorable guarantees. When the difference of pointwise infima can be evaluated or bounded, this decomposition can provide a convenient way to analyze the minimizability gap in terms of the approximation error.

Note that $\mathcal{I}_\ell(\mathcal{H})$ can be non-zero for families of bounded functions. Let $\mathcal{Y} = \{-1, +1\}$ and $\mathcal{H}$ be a family of functions $h$ with $|h(x)| \le \Lambda$ for all $x \in \mathcal{X}$ and such that all values in $[-\Lambda, +\Lambda]$ can be reached. Consider for example the exponential-based margin loss: $\ell(h, x, y) = e^{-yh(x)}$. Let $\eta(x) = \mathsf{p}(+1 \mid x) = \mathcal{D}(Y = +1 \mid X = x)$. Thus, $\mathcal{C}_\ell(h, x) = \eta(x)e^{-h(x)} + (1 - \eta(x))e^{h(x)}$. Then, it is not hard to see that $\mathcal{C}_\ell^*(\mathcal{H}_{\text{all}}, x) = 2\sqrt{\eta(x)(1 - \eta(x))}$ for all $x$ but $\mathcal{C}_\ell^*(\mathcal{H}, x)$ depends on $\Lambda$ with the minimizing value for $h(x)$ being: $\min\{\frac{1}{2} \log \frac{\eta(x)}{1 - \eta(x)}, \Lambda\}$ if $\eta(x) \ge 1/2$, $\max\{\frac{1}{2} \log \frac{\eta(x)}{1 - \eta(x)}, -\Lambda\}$ otherwise. Thus, in the deterministic case, $\mathcal{I}_\ell(\mathcal{H}) = e^{-\Lambda}$.

When the best-in-class error coincides with the Bayes error, $\mathcal{E}_\ell^*(\mathcal{H}) = \mathcal{E}_\ell^*(\mathcal{H}_{\text{all}})$, both the approximation error and minimizability gaps vanish.

**Lemma D.2.** *For any loss function $\ell$ such that $\mathcal{E}_\ell^*(\mathcal{H}) = \mathcal{E}_\ell^*(\mathcal{H}_{\text{all}}) = \mathbb{E}_x[\mathcal{C}_\ell^*(\mathcal{H}_{\text{all}}, x)]$, we have $\mathcal{M}_\ell(\mathcal{H}) = \mathcal{A}_\ell(\mathcal{H}) = 0$.*

*Proof.* By definition, $\mathcal{A}_\ell(\mathcal{H}) = \mathcal{E}_\ell^*(\mathcal{H}) - \mathcal{E}_\ell^*(\mathcal{H}_{\text{all}}) = 0$. Since we have $\mathcal{M}_\ell(\mathcal{H}) \le \mathcal{A}_\ell(\mathcal{H})$, this implies $\mathcal{M}_\ell(\mathcal{H}) = 0$. □

# E  Examples of $\mathcal{H}$-consistency bounds

Here, we compile some common examples of $\mathcal{H}$-consistency bounds for both binary and multi-class classification. Table 1, 2 and 3 include the examples of $\mathcal{H}$-consistency bounds for binary margin-based losses, comp-sum losses and constrained losses, respectively.

These bounds are due to previous work by Awasthi et al. [2022b] for binary margin-based losses, by Mao et al. [2023f] for multi-class comp-sum losses, and by Awasthi et al. [2022a] and Mao et al. [2023b] for multi-class constrained losses, respectively. We consider complete hypothesis sets for binary classification (see Section 4), and symmetric and complete hypothesis sets for multi-class classification (see Section 5).

| $\Phi(u)$ | Comp-sum losses $\ell$ | $\mathcal{H}$-Consistency bounds |
|---|---|---|
| $\frac{1-u}{u}$ | $\sum_{y'\neq y} e^{h(x,y')-h(x,y)}$ | $\mathcal{E}_{\ell_{0-1}}(h) - \mathcal{E}_{\ell_{0-1}}^*(\mathcal{H}) + \mathcal{M}_{\ell_{0-1}}(\mathcal{H}) \leq \sqrt{2}(\mathcal{E}_\ell(h) - \mathcal{E}_\ell^*(\mathcal{H}) + \mathcal{M}_\ell(\mathcal{H}))^{\frac{1}{2}}$ |
| $-\log(u)$ | $-\log\left(\frac{e^{h(x,y)}}{\sum_{y'\in\mathcal{Y}} e^{h(x,y')}}\right)$ | $\mathcal{E}_{\ell_{0-1}}(h) - \mathcal{E}_{\ell_{0-1}}^*(\mathcal{H}) + \mathcal{M}_{\ell_{0-1}}(\mathcal{H}) \leq \sqrt{2}(\mathcal{E}_\ell(h) - \mathcal{E}_\ell^*(\mathcal{H}) + \mathcal{M}_\ell(\mathcal{H}))^{\frac{1}{2}}$ |
| $\frac{1}{\alpha}\left[1 - u^\alpha\right]$ | $\frac{1}{\alpha}\left[1 - \left[\frac{e^{h(x,y)}}{\sum_{y'\in\mathcal{Y}} e^{h(x,y')}}\right]^\alpha\right]$ | $\mathcal{E}_{\ell_{0-1}}(h) - \mathcal{E}_{\ell_{0-1}}^*(\mathcal{H}) + \mathcal{M}_{\ell_{0-1}}(\mathcal{H}) \leq \sqrt{2n^\alpha}(\mathcal{E}_\ell(h) - \mathcal{E}_\ell^*(\mathcal{H}) + \mathcal{M}_\ell(\mathcal{H}))^{\frac{1}{2}}$ |
| $1 - u$ | $1 - \frac{e^{h(x,y)}}{\sum_{y'\in\mathcal{Y}} e^{h(x,y')}}$ | $\mathcal{E}_{\ell_{0-1}}(h) - \mathcal{E}_{\ell_{0-1}}^*(\mathcal{H}) + \mathcal{M}_{\ell_{0-1}}(\mathcal{H}) \leq n(\mathcal{E}_\ell(h) - \mathcal{E}_\ell^*(\mathcal{H}) + \mathcal{M}_\ell(\mathcal{H}))$ |

Table 2: Examples of $\mathcal{H}$-consistency bounds for comp-sum losses.

| $\Phi(u)$ | Constrained losses $\ell$ | $\mathcal{H}$-Consistency bounds |
|---|---|---|
| $e^u$ | $\sum_{y'\neq y} e^{h(x,y')}$ | $\mathcal{E}_{\ell_{0-1}}(h) - \mathcal{E}_{\ell_{0-1}}^*(\mathcal{H}) + \mathcal{M}_{\ell_{0-1}}(\mathcal{H}) \leq \sqrt{2}(\mathcal{E}_\ell(h) - \mathcal{E}_\ell^*(\mathcal{H}) + \mathcal{M}_\ell(\mathcal{H}))^{\frac{1}{2}}$ |
| $\max\{0, 1+u\}^2$ | $\sum_{y'\neq y} \max\{0, 1+h(x,y')\}^2$ | $\mathcal{E}_{\ell_{0-1}}(h) - \mathcal{E}_{\ell_{0-1}}^*(\mathcal{H}) + \mathcal{M}_{\ell_{0-1}}(\mathcal{H}) \leq (\mathcal{E}_\ell(h) - \mathcal{E}_\ell^*(\mathcal{H}) + \mathcal{M}_\ell(\mathcal{H}))^{\frac{1}{2}}$ |
| $(1+u)^2$ | $\sum_{y'\neq y} (1+h(x,y'))^2$ | $\mathcal{E}_{\ell_{0-1}}(h) - \mathcal{E}_{\ell_{0-1}}^*(\mathcal{H}) + \mathcal{M}_{\ell_{0-1}}(\mathcal{H}) \leq (\mathcal{E}_\ell(h) - \mathcal{E}_\ell^*(\mathcal{H}) + \mathcal{M}_\ell(\mathcal{H}))^{\frac{1}{2}}$ |
| $\max\{0, 1+u\}$ | $\sum_{y'\neq y} \max\{0, 1+h(x,y')\}$ | $\mathcal{E}_{\ell_{0-1}}(h) - \mathcal{E}_{\ell_{0-1}}^*(\mathcal{H}) + \mathcal{M}_{\ell_{0-1}}(\mathcal{H}) \leq \mathcal{E}_\ell(h) - \mathcal{E}_\ell^*(\mathcal{H}) + \mathcal{M}_\ell(\mathcal{H})$ |

Table 3: Examples of $\mathcal{H}$-consistency bounds for constrained losses with $\sum_{y\in\mathcal{Y}} h(x,y) = 0$.

## F  Comparison with excess error bounds

Excess error bounds can be used to derive bounds for a hypothesis set $\mathcal{H}$ expressed in terms of the approximation error. Here, we show, however, that, the resulting bounds are looser than $\mathcal{H}$-consistency bounds.

Fix a target loss function $\ell_2$ and a surrogate loss $\ell_1$. Excess error bounds, also known as *surrogate regret bounds*, are bounds relating the excess errors of these loss functions of the following form:

$$\forall h \in \mathcal{H}_{\text{all}}, \quad \psi\big(\mathcal{E}_{\ell_2}(h) - \mathcal{E}_{\ell_2}^*(\mathcal{H}_{\text{all}})\big) \leq \mathcal{E}_{\ell_1}(h) - \mathcal{E}_{\ell_1}^*(\mathcal{H}_{\text{all}}), \tag{8}$$

where $\psi:\mathbb{R}_+ \to \mathbb{R}_+$ is a non-decreasing and convex function on $\mathbb{R}_+$. Recall that as shown in (1), the excess error can be written as the sum of the estimation error and the approximation error. Thus, the excess error bound can be equivalently expressed as follows:

$$\forall h \in \mathcal{H}_{\text{all}}, \quad \psi\big(\mathcal{E}_{\ell_2}(h) - \mathcal{E}_{\ell_2}^*(\mathcal{H}) + \mathcal{A}_{\ell_2}(\mathcal{H})\big) \leq \mathcal{E}_{\ell_1}(h) - \mathcal{E}_{\ell_1}^*(\mathcal{H}) + \mathcal{A}_{\ell_1}(\mathcal{H}). \tag{9}$$

In Section 3, we have shown that the minimizabiliy gap can be upper bounded by the approximation error $\mathcal{M}_\ell(\mathcal{H}) \leq \mathcal{A}(\mathcal{H})$ and is in general a finer quantity for a surrogate loss $\ell_1$. However, we will show that for a target loss $\ell_2$ that is *discrete*, the minimizabiliy gap in general coincides with the approximation error.

**Definition F.1.** We say that a target loss $\ell_2$ is *discrete* if we can write $\ell_2(h, x, y) = \mathsf{L}(\mathsf{h}(x), y)$ for some binary function $\mathsf{L}:\mathcal{Y} \times \mathcal{Y} \to \mathbb{R}_+$.

In other words, a discrete target loss $\ell_2$ is explicitly a function of both the prediction $\mathsf{h}(x)$ and the true label $y$, where both belong to the label space $\mathcal{Y}$. Consequently, it can assume at most $n^2$ distinct discrete values.

Next, we demonstrate that for such discrete target loss functions, if for any instance, the set of predictions generated by the hypothesis set completely spans the label space, then the minimizability gap is precisely equal to the approximation error. For convenience, we denote by $\mathsf{H}(x)$ the set of predictions generated by the hypothesis set on input $x \in \mathcal{X}$, defined as $\mathsf{H}(x) = \{\mathsf{h}(x): h \in \mathcal{H}\}$

**Theorem F.2.** *Given a discrete target loss function $\ell_2$. Assume that the hypothesis set $\mathcal{H}$ satisfies, for any $x \in \mathcal{X}$, $\mathsf{H}(x) = \mathcal{Y}$. Then, we have $\mathcal{I}_{\ell_2}(\mathcal{H}) = 0$ and $\mathcal{M}_{\ell_2}(\mathcal{H}) = \mathcal{A}_{\ell_2}(\mathcal{H})$.*

*Proof.* As shown in Section 3, the minimizability gap can be decomposed in terms of the approximation error and the difference of pointwise infima:

$$\mathcal{M}_{\ell_2}(\mathcal{H}) = \mathcal{A}_{\ell_2}(\mathcal{H}) - \mathcal{I}_{\ell_2}(\mathcal{H})$$
$$= \mathcal{A}_{\ell_2}(\mathcal{H}) - \mathbb{E}_x\Big[\mathcal{C}_{\ell_2}^*(\mathcal{H}, x) - \mathcal{C}_{\ell_2}^*(\mathcal{H}_{\text{all}}, x)\Big].$$

By definition and the fact that $\ell_2$ is discrete, the conditional error can be written as

$$\mathcal{C}_{\ell_2}(h, x) = \sum_{y \in \mathcal{Y}} \mathsf{p}(y|x)\ell_2(h, x, y) = \sum_{y \in \mathcal{Y}} \mathsf{p}(y|x)\mathsf{L}(\mathsf{h}(x), y).$$

Thus, for any $x \in \mathcal{X}$, the best-in-class conditional error can be expressed as

$$\mathcal{C}_{\ell_2}^*(\mathcal{H}, x) = \inf_{h \in \mathcal{H}} \sum_{y \in \mathcal{Y}} \mathsf{p}(y|x)\mathsf{L}(\mathsf{h}(x), y) = \inf_{y' \in \mathsf{H}(x)} \sum_{y \in \mathcal{Y}} \mathsf{p}(y|x)\mathsf{L}(y', y).$$

By the assumption that $\mathsf{H}(x) = \mathcal{Y}$, we obtain

$$\forall x \in \mathcal{X}, \quad \mathcal{C}_{\ell_2}^*(\mathcal{H}, x) = \inf_{y' \in \mathsf{H}(x)} \sum_{y \in \mathcal{Y}} \mathsf{p}(y|x)\mathsf{L}(y', y) = \inf_{y' \in \mathcal{Y}} \sum_{y \in \mathcal{Y}} \mathsf{p}(y|x)\mathsf{L}(y', y) = \mathcal{C}_{\ell_2}^*(\mathcal{H}_{\mathrm{all}}, x).$$

Therefore, $\mathcal{I}_{\ell_2}(\mathcal{H}) = \mathbb{E}_x\Big[\mathcal{C}_{\ell_2}^*(\mathcal{H}, x) - \mathcal{C}_{\ell_2}^*(\mathcal{H}_{\mathrm{all}}, x)\Big] = 0$ and $\mathcal{M}_{\ell_2}(\mathcal{H}) = \mathcal{A}_{\ell_2}(\mathcal{H})$. $\qquad \square$

By Theorem F.2, for a target loss $\ell_2$ that is discrete and hypothesis sets $\mathcal{H}$ modulo mild assumptions, the minimizabiliy gap coincides with the approximation error. In such cases, by comparing an excess error bound (9) with the $\mathcal{H}$-consistency bound (6):

Excess error bound: $\quad \psi\big(\mathcal{E}_{\ell_2}(h) - \mathcal{E}_{\ell_2}^*(\mathcal{H}) + \mathcal{A}_{\ell_2}(\mathcal{H})\big) \leq \mathcal{E}_{\ell_1}(h) - \mathcal{E}_{\ell_1}^*(\mathcal{H}) + \mathcal{A}_{\ell_1}(\mathcal{H})$

$\mathcal{H}$-consistency bound: $\quad \psi\big(\mathcal{E}_{\ell_2}(h) - \mathcal{E}_{\ell_2}^*(\mathcal{H}) + \mathcal{M}_{\ell_2}(\mathcal{H})\big) \leq \mathcal{E}_{\ell_1}(h) - \mathcal{E}_{\ell_1}^*(\mathcal{H}) + \mathcal{M}_{\ell_1}(\mathcal{H}),$

we obtain that the left-hand side of both bounds are equal (since $\mathcal{M}_{\ell_2}(\mathcal{H}) = \mathcal{A}_{\ell_2}(\mathcal{H})$), while the right-hand side of the $\mathcal{H}$-consistency bound is always upper bounded by and can be finer than the right-hand side of the excess error bound (since $\mathcal{M}_{\ell_1}(\mathcal{H}) \leq \mathcal{A}_{\ell_1}(\mathcal{H})$), which implies that excess error bounds (or surrogate regret bounds) are in general inferior to $\mathcal{H}$-consistency bounds.

## G  Polyhedral losses versus smooth losses

Since $\mathcal{H}$-consistency bounds subsume excess error bounds as a special case (Appendix F), the linear growth rate of polyhedral loss excess error bounds (Finocchiaro et al. [2019]) also dictates a linear growth rate for polyhedral $\mathcal{H}$-consistency bounds, if they exist. This is illustrated by the hinge loss or $\rho$-margin loss which have been shown to benefit from $\mathcal{H}$-consistency bounds [Awasthi et al., 2022b].

Here, we compare in more detail polyhedral losses and the smooth losses. Assume that a hypothesis set $\mathcal{H}$ is complete and thus $\mathsf{H}(x) = \mathcal{Y}$ for any $x \in \mathcal{X}$. By Theorem F.2, we have $\mathcal{A}_{\ell_{0-1}}(\mathcal{H}) = \mathcal{M}_{\ell_{0-1}}(\mathcal{H})$. As shown by Frongillo and Waggoner [2021, Theorem 3], a Bayes-consistent polyhedral loss $\Phi_{\mathrm{poly}}$ admits the following linear excess error bound, for some $\beta_1 > 0$,

$$\forall h \in \mathcal{H}, \ \beta_1\big(\mathcal{E}_{\ell_{0-1}}(h) - \mathcal{E}_{\ell_{0-1}}^*(\mathcal{H}) + \mathcal{M}_{\ell_{0-1}}(\mathcal{H})\big) \leq \mathcal{E}_{\Phi_{\mathrm{poly}}}(h) - \mathcal{E}_{\Phi_{\mathrm{poly}}}^*(\mathcal{H}) + \mathcal{A}_{\Phi_{\mathrm{poly}}}(\mathcal{H}). \quad (10)$$

However, for a smooth loss $\Phi_{\mathrm{smooth}}$, if it satisfies the condition of Theorem 4.2, $\Phi_{\mathrm{smooth}}$ admits the following $\mathcal{H}$-consistency bound:

$$\forall h \in \mathcal{H}, \ \mathcal{T}\big(\mathcal{E}_{\ell_{0-1}}(h) - \mathcal{E}_{\ell_{0-1}}^*(\mathcal{H}) + \mathcal{M}_{\ell_{0-1}}(\mathcal{H})\big) \leq \mathcal{E}_{\Phi_{\mathrm{smooth}}}(h) - \mathcal{E}_{\Phi_{\mathrm{smooth}}}^*(\mathcal{H}) + \mathcal{M}_{\Phi_{\mathrm{smooth}}}(\mathcal{H}). \quad (11)$$

where $\mathcal{T}(t) = \Theta(t^2)$. Therefore, our theory offers a principled basis for comparing polyhedral losses (10) and smooth losses (11), which depends on the following factors:

- The growth rate: linear for polyhedral losses, while square-root for smooth losses.

- The optimization property: smooth losses are more favorable for optimization compared to polyhedral losses, in particular with deep neural networks.

- The approximation theory: the approximation error $\mathcal{A}_{\Phi_{\mathrm{poly}}}(\mathcal{H})$ appears on the right-hand side of the bound for polyhedral losses, whereas a finer quantity, the minimizability gap $\mathcal{M}_{\Phi_{\mathrm{smooth}}}(\mathcal{H})$, is present on the right-hand side of the bound for smooth losses.

# H Comparison of minimizability gaps across comp-sum losses

For $\ell_\tau^{\mathrm{comp}}$ loss functions, $\tau \in [0,2)$, we can characterize minimizability gaps as follows.

**Theorem H.1.** *Assume that for any $x \in \mathcal{X}$, we have $\{(h(x,1),\ldots,h(x,n)) : h \in \mathcal{H}\} = [-\Lambda,+\Lambda]^n$. Then, for comp-sum losses $\ell_\tau^{\mathrm{comp}}$ and any deterministic distribution, the minimizability gaps can be expressed as follows:*

$$\mathcal{M}_{\ell_\tau^{\mathrm{comp}}}(\mathcal{H}) \le \widetilde{\mathcal{M}}_{\ell_\tau^{\mathrm{comp}}}(\mathcal{H}) = f_\tau\Big(\mathcal{R}^*_{\ell_{\tau=0}^{\mathrm{comp}}}(\mathcal{H})\Big) - f_\tau\Big(\mathcal{C}^*_{\ell_{\tau=0}^{\mathrm{comp}}}(\mathcal{H},x)\Big), \tag{12}$$

*where $f_\tau(u) = \log(1+u)1_{\tau=1} + \frac{1}{1-\tau}\big((1+u)^{1-\tau}-1\big)1_{\tau\ne1}$ and $\mathcal{C}^*_{\ell_{\tau=0}^{\mathrm{comp}}}(\mathcal{H},x) = e^{-2\Lambda}(n-1)$. Moreover, $\widetilde{\mathcal{M}}_{\ell_\tau^{\mathrm{comp}}}(\mathcal{H})$ is a non-increasing function of $\tau$.*

*Proof.* Since $f_\tau$ is concave and non-decreasing, and the equality $\ell_\tau = f_\tau(\ell_{\tau=0})$ holds, the minimizability gaps can be upper bounded as follows, for any $\tau \ge 0$,

$$\mathcal{M}_{\ell_\tau^{\mathrm{comp}}}(\mathcal{H}) \le f_\tau\Big(\mathcal{R}^*_{\ell_{\tau=0}^{\mathrm{comp}}}(\mathcal{H})\Big) - \mathbb{E}_x\big[\mathcal{C}^*_{\ell_\tau^{\mathrm{comp}}}(\mathcal{H},x)\big].$$

Since the distribution is deterministic, the conditional error can be expressed as follows:

$$\mathcal{C}_{\ell_\tau^{\mathrm{comp}}}(h,x) = f_\tau\left(\sum_{y'\ne y_{\max}} \exp(h(x,y') - h(x,y_{\max}))\right) \tag{13}$$

where $y_{\max} = \mathrm{argmax}\, \mathsf{p}(y\,|\,x)$. Using the fact that $f_\tau$ is increasing for any $\tau > 0$, the hypothesis $h^* \colon (x,y) \mapsto \Lambda 1_{y=y_{\max}} - \Lambda 1_{y\ne y_{\max}}$ achieves the best-in-class conditional error. Thus,

$$\mathcal{C}^*_{\ell_\tau^{\mathrm{comp}}}(\mathcal{H},x) = \mathcal{C}_{\ell_\tau^{\mathrm{comp}}}(h^*,x) = f_\tau\Big(\mathcal{C}^*_{\ell_{\tau=0}^{\mathrm{comp}}}(\mathcal{H},x)\Big)$$

where $\mathcal{C}^*_{\ell_{\tau=0}^{\mathrm{comp}}}(\mathcal{H},x) = e^{-2\Lambda}(n-1)$. Therefore,

$$\mathcal{M}_{\ell_\tau^{\mathrm{comp}}}(\mathcal{H}) \le f_\tau\Big(\mathcal{R}^*_{\ell_{\tau=0}^{\mathrm{comp}}}(\mathcal{H})\Big) - f_\tau\Big(\mathcal{C}^*_{\ell_{\tau=0}^{\mathrm{comp}}}(\mathcal{H},x)\Big).$$

This completes the first part of the proof. Using the fact that $\tau \mapsto f_\tau(u_1) - f_\tau(u_2)$ is a non-increasing function of $\tau$ for any $u_1 \ge u_2 \ge 0$, the second proof is completed as well. $\square$

The theorem shows that for comp-sum loss functions $\ell_\tau^{\mathrm{comp}}$, the minimizability gaps are non-increasing with respect to $\tau$. Note that $\Phi^\tau$ satisfies the conditions of Theorem 5.3 for any $\tau \in [0,2)$. Therefore, focusing on behavior near zero (ignoring constants), the theorem provides a principled comparison of minimizability gaps and $\mathcal{H}$-consistency bounds across different comp-sum losses.

# I Small surrogate minimizability gaps: proof for binary classification

**Theorem 6.1.** *Assume that $\mathcal{D}$ is deterministic and that the best-in-class error is achieved by some $h^* \in \mathcal{H}$. Then, the minimizability gap is null, $\mathcal{M}(\mathcal{H}) = 0$, iff*

$$\ell(h^*(x),+1) = \ell_+ \text{ a.s. over } \mathcal{X}_+, \quad \ell(h^*(x),-1) = \ell_- \text{ a.s. over } \mathcal{X}_-.$$

*If further $\alpha \mapsto \ell(\alpha,+1)$ and $\alpha \mapsto \ell(\alpha,-1)$ are injective and $\ell_+ = \ell(\alpha_+,+1)$, $\ell_- = \ell(\alpha_-,-1)$, then, the condition is equivalent to $h^*(x) = \alpha_+ 1_{x\in\mathcal{X}_+} + \alpha_- 1_{x\in\mathcal{X}_-}$. Furthermore, the minimizability gap is bounded by $\epsilon$ iff $p(\mathbb{E}[\ell(h^*(x),+1)\,|\,y=+1] - \ell_+) + (1-p)(\mathbb{E}[\ell(h^*(x),-1)\,|\,y=-1] - \ell_-) \le \epsilon$. In particular, the condition implies:*

$$\mathbb{E}[\ell(h^*(x),+1)\,|\,y=+1] - \ell_+ \le \frac{\epsilon}{p} \quad and \quad \mathbb{E}[\ell(h^*(x),-1)\,|\,y=-1] - \ell_- \le \frac{\epsilon}{1-p}.$$

*Proof.* By definition of $h^*$, using the shorthand $p = \mathbb{P}[y=+1]$, we can write

$$\inf_{h\in\mathcal{H}} \mathbb{E}[\ell(h(x),y)] = \mathbb{E}[\ell(h^*(x),y)]$$
$$= p\,\mathbb{E}[\ell(h^*(x),+1)\,|\,y=+1] + (1-p)\,\mathbb{E}[\ell(h^*(x),-1)\,|\,y=-1].$$

Since the distribution is deterministic, the expected pointwise infimum can be rewritten as follows:

$$\mathbb{E}_x\left[\inf_{h\in\mathcal{H}}\mathbb{E}_y[\ell(h(x),y)\mid x]\right]=\mathbb{E}_x\left[\inf_{\alpha\in A}\mathbb{E}_y[\ell(\alpha,y)\mid x]\right]=p\inf_{\alpha\in A}\ell(\alpha,+1)+(1-p)\inf_{\alpha\in A}\ell(\alpha,-1)$$
$$=p\ell_++(1-p)\ell_-,$$

where $\ell_+=\inf_{\alpha\in A}\ell(\alpha,+1)$ and $\ell_-=\inf_{\alpha\in A}\ell(\alpha,-1)$. Thus, we have

$$\mathcal{M}(\mathcal{H})=p\,\mathbb{E}[\ell(h^*(x),+1)-\ell_+\mid y=+1]+(1-p)\,\mathbb{E}[\ell(h^*(x),-1)-\ell_-\mid y=-1].$$

In view of that, since, by definition of $\ell_+$ and $\ell_-$, the expressions within the conditional expectations are non-negative, the equality $\mathcal{M}(\mathcal{H})=0$ holds iff $\ell(h^*(x),+1)-\ell_+=0$ almost surely for any $x$ in $\mathcal{X}_+$ and $\ell(h^*(x),-1)-\ell_-=0$ almost surely for any $x$ in $\mathcal{X}_-$. This completes the first part of the proof. Furthermore, $\mathcal{M}(\mathcal{H})\le\epsilon$ is equivalent to

$$p\,\mathbb{E}[\ell(h^*(x),+1)-\ell_+\mid y=+1]+(1-p)\,\mathbb{E}[\ell(h^*(x),-1)-\ell_-\mid y=-1]\le\epsilon$$

that is

$$p(\mathbb{E}[\ell(h^*(x),+1)\mid y=+1]-\ell_+)+(1-p)(\mathbb{E}[\ell(h^*(x),-1)\mid y=-1]-\ell_-)\le\epsilon.$$

In light of the non-negativity of the expressions, this implies in particular:

$$\mathbb{E}[\ell(h^*(x),+1)\mid y=+1]-\ell_+\le\frac{\epsilon}{p}\quad\text{and}\quad\mathbb{E}[\ell(h^*(x),-1)\mid y=-1]-\ell_-\le\frac{\epsilon}{1-p}.$$

This completes the second part of the proof. $\qquad\square$

**Theorem 6.2.** *The best-in-class error is achieved by some $h^*\in\mathcal{H}$ and the minimizability gap is null, $\mathcal{M}(\mathcal{H})=0$, iff there exists $h^*\in\mathcal{H}$ such that for all $x$,*

$$\mathbb{E}_y[\ell(h^*(x),y)\mid x]=\inf_{\alpha\in A}\mathbb{E}_y[\ell(\alpha,y)\mid x]\text{ a.s. over }\mathcal{X}. \tag{3}$$

*If further $\alpha\mapsto\mathbb{E}_y[\ell(\alpha,y)\mid x]$ is injective and $\inf_{\alpha\in A}\mathbb{E}_y[\ell(\alpha,y)\mid x]=\mathbb{E}_y[\ell(\alpha^*(x),y)\mid x]$, then, the condition is equivalent to $h^*(x)=\alpha^*(x)$ a.s. for $x\in\mathcal{X}$. Furthermore, the minimizability gap is bounded by $\epsilon$, $\mathcal{M}(\mathcal{H})\le\epsilon$, iff there exists $h^*\in\mathcal{H}$ such that*

$$\mathbb{E}_x\left[\mathbb{E}_y[\ell(h^*(x),y)\mid x]-\inf_{\alpha\in A}\mathbb{E}_y[\ell(\alpha,y)\mid x]\right]\le\epsilon. \tag{4}$$

*Proof.* Assume that the best-in-class error is achieved by some $h^*\in\mathcal{H}$. Then, we can write

$$\inf_{h\in\mathcal{H}}\mathbb{E}[\ell(h(x),y)]=\mathbb{E}[\ell(h^*(x),y)]=\mathbb{E}_x\left[\mathbb{E}_y[\ell(h^*(x),y)\mid x]\right].$$

The expected pointwise infimum can be rewritten as follows:

$$\mathbb{E}_x\left[\inf_{h\in\mathcal{H}}\mathbb{E}_y[\ell(h(x),y)\mid x]\right]=\mathbb{E}_x\left[\inf_{\alpha\in A}\mathbb{E}_y[\ell(\alpha,y)\mid x]\right].$$

Thus, we have

$$\mathcal{M}(\mathcal{H})=\mathbb{E}_x\left[\mathbb{E}_y[\ell(h^*(x),y)\mid x]-\inf_{\alpha\in A}\mathbb{E}_y[\ell(\alpha,y)\mid x]\right].$$

In view of that, since, by the definition of infimum, the expressions within the marginal expectations are non-negative, the condition that $\mathcal{M}(\mathcal{H})=0$ implies that

$$\mathbb{E}_y[\ell(h^*(x),y)\mid x]=\inf_{\alpha\in A}\mathbb{E}_y[\ell(\alpha,y)\mid x]\text{ a.s. over }\mathcal{X}.$$

On the other hand, if there exists $h^*\in\mathcal{H}$ such that the condition (3) holds, then,

$$\mathcal{M}(\mathcal{H})=\inf_{h\in\mathcal{H}}\mathbb{E}[\ell(h(x),y)]-\mathbb{E}_x\left[\inf_{\alpha\in A}\mathbb{E}_y[\ell(\alpha,y)\mid x]\right]\le\mathbb{E}_x\left[\mathbb{E}_y[\ell(h^*(x),y)\mid x]-\inf_{\alpha\in A}\mathbb{E}_y[\ell(\alpha,y)\mid x]\right]=0.$$

Since $\mathcal{M}(\mathcal{H})$ is non-negative, the inequality is achieved. Thus, we have

$$\mathcal{M}(\mathcal{H})=0\text{ and }\inf_{h\in\mathcal{H}}\mathbb{E}[\ell(h(x),y)]=\mathbb{E}[\ell(h^*(x),y)].$$

If there exists $h^* \in \mathcal{H}$ such that the condition (4) holds, then,

$$\mathcal{M}(\mathcal{H}) = \inf_{h \in \mathcal{H}} \mathbb{E}[\ell(h(x), y)] - \mathbb{E}_x\left[\inf_{\alpha \in A} \mathbb{E}_y[\ell(\alpha, y) \mid x]\right] \leq \mathbb{E}_x\left[\mathbb{E}_y[\ell(h^*(x), y) \mid x] - \inf_{\alpha \in A} \mathbb{E}_y[\ell(\alpha, y) \mid x]\right] = \epsilon.$$

On the other hand, since we have

$$\mathcal{M}(\mathcal{H}) = \mathbb{E}_x\left[\mathbb{E}_y[\ell(h^*(x), y) \mid x] - \inf_{\alpha \in A} \mathbb{E}_y[\ell(\alpha, y) \mid x]\right],$$

$\mathcal{M}(\mathcal{H}) \leq \epsilon$ implies that

$$\mathbb{E}_x\left[\mathbb{E}_y[\ell(h^*(x), y) \mid x] - \inf_{\alpha \in A} \mathbb{E}_y[\ell(\alpha, y) \mid x]\right] \leq \epsilon.$$

This completes the proof. $\qquad\qquad\square$

## J    Small surrogate minimizability gaps: multi-class classification

We consider the multi-class setting with label space $[n] = \{1, 2, \ldots, n\}$. In this setting, the surrogate loss incurred by a predictor $h$ at a labeled point $(x, y)$ can be expressed by $\ell(h(x), y)$, where $h(x) = [h(x, 1), \ldots, h(x, n)]$ is the score vector of the predictor $h$. We denote by $A$ the set of values in $\mathbb{R}^n$ taken by the score vector of predictors in $\mathcal{H}$ at $x$, which we assume to be independent of $x$: $A = \{h(x) : h \in \mathcal{H}\}$, for all $x \in \mathcal{X}$.

### J.1    Deterministic scenario

We first consider the deterministic scenario, where the conditional probability $\mathsf{p}(y \mid x)$ is either zero or one. For a deterministic distribution, we denote by $\mathcal{X}_k$ the subset of $\mathcal{X}$ over which the label is $k$. For convenience, let $\ell_k = \inf_{\alpha \in A} \ell(\alpha, k)$, for any $k \in [n]$.

**Theorem J.1.** *Assume that $\mathcal{D}$ is deterministic and that the best-in-class error is achieved by some $h^* \in \mathcal{H}$. Then, the minimizability gap is null, $\mathcal{M}(\mathcal{H}) = 0$, iff*

$$\forall k \in [n],\ \ell(h^*(x), k) = \ell_k \text{ a.s. over } \mathcal{X}_k.$$

*If further $\alpha \mapsto \ell(\alpha, k)$ is injective and $\ell_k = \ell(\alpha_k, k)$ for all $k \in [n]$, then, the condition is equivalent to $\forall k \in [n], h^*(x) = \alpha_k$ a.s. for $x \in \mathcal{X}_k$. Furthermore, the minimizability gap is bounded by $\epsilon$, $\mathcal{M}(\mathcal{H}) \leq \epsilon$, iff*

$$\sum_{k \in [n]} p_k(\mathbb{E}[\ell(h^*(x), k) \mid y = k] - \ell_k) \leq \epsilon.$$

*In particular, the condition implies:*

$$\mathbb{E}[\ell(h^*(x), k) \mid y = k] - \ell_k \leq \frac{\epsilon}{p_k},\ \forall k \in [n],$$

*Proof.* By definition of $h^*$, using the shorthand $p_k = \mathbb{P}[y = k]$ for any $k \in [n]$, we can write

$$\inf_{h \in \mathcal{H}} \mathbb{E}[\ell(h(x), y)] = \mathbb{E}[\ell(h^*(x), y)] = \sum_{k \in [n]} p_k \mathbb{E}[\ell(h^*(x), k) \mid y = k].$$

Since the distribution is deterministic, the expected pointwise infimum can be rewritten as follows:

$$\mathbb{E}_x\left[\inf_{h \in \mathcal{H}} \mathbb{E}_y[\ell(h(x), y) \mid x]\right] = \mathbb{E}_x\left[\inf_{\alpha \in A} \mathbb{E}_y[\ell(\alpha, y) \mid x]\right] = \sum_{k \in [n]} p_k \inf_{\alpha \in A} \ell(\alpha, k) = \sum_{k \in [n]} p_k \ell_k,$$

where $\ell_k = \inf_{\alpha \in A} \ell(\alpha, k)$, for any $k \in [n]$. Thus, we have

$$\mathcal{M}(\mathcal{H}) = \sum_{k \in [n]} p_k \mathbb{E}[\ell(h^*(x), k) - \ell_k \mid y = k].$$

In view of that, since, by definition of $\ell_k$, the expressions within the conditional expectations are non-negative, the equality $\mathcal{M}(\mathcal{H}) = 0$ holds iff $\ell(h^*(x), k) - \ell_k = 0$ almost surely for any $x$ in $\mathcal{X}_k$, $\forall k \in [n]$. Furthermore, $\mathcal{M}(\mathcal{H}) \leq \epsilon$ is equivalent to

$$\sum_{k \in [n]} p_k \mathbb{E}[\ell(h^*(x), k) - \ell_k \mid y = k] \leq \epsilon$$

that is

$$\sum_{k \in [n]} p_k \left( \mathbb{E}[\ell(h^*(x), k) \mid y = k] - \ell_k \right) \le \epsilon.$$

In light of the non-negativity of the expressions, this implies in particular:

$$\mathbb{E}[\ell(h^*(x), k) \mid y = k] - \ell_k \le \frac{\epsilon}{p_k}, \ \forall k \in [n].$$

This completes the proof. □

The theorem suggests that, under those assumptions, for the surrogate minimizability gap to be zero, the score vector of best-in-class hypothesis must be piecewise constant with specific values on $\mathcal{X}_k$. The existence of such a hypothesis in $\mathcal{H}$ depends both on the complexity of the decision surface separating $\mathcal{X}_k$ and on that of the hypothesis set $\mathcal{H}$. The theorem also suggests that when the score vector of best-in-class classifier $\epsilon$-approximates $\alpha_k$ over $\mathcal{X}_k$ for any $k \in [n]$, then the minimizability gap is bounded by $\epsilon$. The existence of such a hypothesis in $\mathcal{H}$ depends on the complexity of the decision surface.

### J.2 Stochastic scenario

In the previous sections, we analyze instances featuring small minimizability gaps in a deterministic setting. Moving forward, we aim to extend this analysis to the stochastic scenario. We first provide two general results, which are the direct extensions of that in the deterministic scenario. The following result shows that the minimizability gap is zero when there exists $h^* \in \mathcal{H}$ that matches $\alpha^*(x)$ for all $x$, where $\alpha^*(x)$ is the minimizer of the conditional error. It also shows that the minimizability gap is bounded by $\epsilon$ when there exists $h^* \in \mathcal{H}$ whose conditional error $\epsilon$-approximates best-in-class conditional error for all $x$.

**Theorem J.2.** *The best-in-class error is achieved by some $h^* \in \mathcal{H}$ and the minimizability gap is null, $\mathcal{M}(\mathcal{H}) = 0$, iff there exists $h^* \in \mathcal{H}$ such that*

$$\mathbb{E}_y[\ell(h^*(x), y) \mid x] = \inf_{\alpha \in A} \mathbb{E}_y[\ell(\alpha, y) \mid x] \ a.s. \ over \ \mathcal{X}. \tag{14}$$

*If further $\alpha \mapsto \mathbb{E}_y[\ell(\alpha, y) \mid x]$ is injective and $\inf_{\alpha \in A} \mathbb{E}_y[\ell(\alpha, y) \mid x] = \mathbb{E}_y[\ell(\alpha^*(x), y) \mid x]$, then, the condition is equivalent to $h^*(x) = \alpha^*(x)$ a.s. for $x \in \mathcal{X}$. Furthermore, the minimizability gap is bounded by $\epsilon$, $\mathcal{M}(\mathcal{H}) \le \epsilon$, iff there exists $h^* \in \mathcal{H}$ such that*

$$\mathbb{E}_x\left[ \mathbb{E}_y[\ell(h^*(x), y) \mid x] - \inf_{\alpha \in A} \mathbb{E}_y[\ell(\alpha, y) \mid x] \right] \le \epsilon. \tag{15}$$

*Proof.* Assume that the best-in-class error is achieved by some $h^* \in \mathcal{H}$. Then, we can write

$$\inf_{h \in \mathcal{H}} \mathbb{E}[\ell(h(x), y)] = \mathbb{E}[\ell(h^*(x), y)] = \mathbb{E}_x\left[ \mathbb{E}_y[\ell(h^*(x), y) \mid x] \right].$$

The expected pointwise infimum can be rewritten as follows:

$$\mathbb{E}_x\left[ \inf_{h \in \mathcal{H}} \mathbb{E}_y[\ell(h(x), y) \mid x] \right] = \mathbb{E}_x\left[ \inf_{\alpha \in A} \mathbb{E}_y[\ell(\alpha, y) \mid x] \right].$$

Thus, we have

$$\mathcal{M}(\mathcal{H}) = \mathbb{E}_x\left[ \mathbb{E}_y[\ell(h^*(x), y) \mid x] - \inf_{\alpha \in A} \mathbb{E}_y[\ell(\alpha, y) \mid x] \right].$$

In view of that, since, by the definition of infimum, the expressions within the marginal expectations are non-negative, the condition that $\mathcal{M}(\mathcal{H}) = 0$ implies that

$$\mathbb{E}_y[\ell(h^*(x), y) \mid x] = \inf_{\alpha \in A} \mathbb{E}_y[\ell(\alpha, y) \mid x] \ a.s. \ over \ \mathcal{X}.$$

On the other hand, if there exists $h^* \in \mathcal{H}$ such that the condition (3) holds, then,

$$\mathcal{M}(\mathcal{H}) = \inf_{h \in \mathcal{H}} \mathbb{E}[\ell(h(x), y)] - \mathbb{E}_x\left[ \inf_{\alpha \in A} \mathbb{E}_y[\ell(\alpha, y) \mid x] \right] \le \mathbb{E}_x\left[ \mathbb{E}_y[\ell(h^*(x), y) \mid x] - \inf_{\alpha \in A} \mathbb{E}_y[\ell(\alpha, y) \mid x] \right] = 0.$$

Since $\mathcal{M}(\mathcal{H})$ is non-negative, the inequality is achieved. Thus, we have

$$\mathcal{M}(\mathcal{H}) = 0 \text{ and } \inf_{h \in \mathcal{H}} \mathbb{E}[\ell(h(x), y)] = \mathbb{E}[\ell(h^*(x), y)].$$

If there exists $h^* \in \mathcal{H}$ such that the condition (4) holds, then,

$$\mathcal{M}(\mathcal{H}) = \inf_{h \in \mathcal{H}} \mathbb{E}[\ell(h(x), y)] - \mathbb{E}_x\Big[\inf_{\alpha \in A} \mathbb{E}_y[\ell(\alpha, y) \mid x]\Big] \le \mathbb{E}_x\Big[\mathbb{E}_y[\ell(h^*(x), y) \mid x] - \inf_{\alpha \in A} \mathbb{E}_y[\ell(\alpha, y) \mid x]\Big] = \epsilon.$$

On the other hand, since we have

$$\mathcal{M}(\mathcal{H}) = \mathbb{E}_x\Big[\mathbb{E}_y[\ell(h^*(x), y) \mid x] - \inf_{\alpha \in A} \mathbb{E}_y[\ell(\alpha, y) \mid x]\Big],$$

$\mathcal{M}(\mathcal{H}) \le \epsilon$ implies that

$$\mathbb{E}_x\Big[\mathbb{E}_y[\ell(h^*(x), y) \mid x] - \inf_{\alpha \in A} \mathbb{E}_y[\ell(\alpha, y) \mid x]\Big] \le \epsilon.$$

This completes the proof. $\qquad\square$

### J.3 Examples

Note that when the distribution is assumed to be deterministic, the condition (4) and condition (15) are reduced to the condition of Theorem 6.1 in binary classification and that of Theorem J.1 in multi-class classification, respectively. In the stochastic scenario, the existence of such a hypothesis not only depends on the complexity of the decision surface, but also depends on the distributional assumption on the conditional distribution $p(x) = (\mathsf{p}(y|x))_{y \in \mathcal{Y}}$, where $\mathsf{p}(y|x) = \mathcal{D}(Y = y | X = x)$ is the conditional probability of $Y = y$ given $X = x$. In the binary classification, we have $p(x) = (\mathsf{p}(+1|x), \mathsf{p}(-1|x))$, where $\mathsf{p}(+1 \mid x) + \mathsf{p}(-1 \mid x) = 1$. For simplicity, we use the notation $\eta(x)$ and $1 - \eta(x)$ to represent $\mathsf{p}(+1|x)$ and $\mathsf{p}(-1|x)$ respectively. In the multi-class classification with $\mathcal{Y} = \{1, \ldots, n\}$, we have $p(x) = (\mathsf{p}(1|x), \mathsf{p}(2|x), \ldots, \mathsf{p}(n|x))$ where $n$ is the number of classes. As examples, here too, we examine exponential loss and logistic loss in binary classification and multi-class logistic loss in multi-class classification.

**A. Example: binary classification.** Let $\epsilon \in [0, \frac{1}{2}]$. We denote by $\mathcal{X}_+$ the subset of $\mathcal{X}$ over which $\eta(x) = 1 - \epsilon$ and by $\mathcal{X}_-$ the subset of $\mathcal{X}$ over which $\eta(x) = \epsilon$. Let $\mathcal{H}$ be a family of functions $h$ with $|h(x)| \le \Lambda$ for all $x \in \mathcal{X}$ and such that all values in $[-\Lambda, +\Lambda]$ can be reached. Thus, $A = [-\Lambda, \Lambda]$ for any $x \in \mathcal{X}$. Consider the exponential loss: $\ell(h, x, y) = e^{-yh(x)}$. Then, for any $x \in \mathcal{X}$ and $\alpha \in A$, we have

$$\mathbb{E}_y[\ell(\alpha, y) \mid x] = \begin{cases} (1 - \epsilon)e^{-\alpha} + \epsilon e^{\alpha} & x \in \mathcal{X}_+ \\ \epsilon e^{-\alpha} + (1 - \epsilon)e^{\alpha} & x \in \mathcal{X}_-. \end{cases}$$

Thus, it is not hard to see that for any $\epsilon \le \frac{1}{e^{2\Lambda}+1}$, the infimum $\inf_{\alpha \in A} \mathbb{E}_y[\ell(\alpha, y) \mid x]$ can be achieved by $\alpha^*(x) = \begin{cases} \Lambda & x \in \mathcal{X}_+ \\ -\Lambda & x \in \mathcal{X}_- \end{cases} \in A$. Similarly, for the logistic loss $\ell(h, x, y) = \log(1 + e^{-yh(x)})$, we have that

$$\mathbb{E}_y[\ell(\alpha, y) \mid x] = \begin{cases} (1 - \epsilon)\log(1 + e^{-\alpha}) + \epsilon \log(1 + e^{\alpha}) & x \in \mathcal{X}_+ \\ \epsilon \log(1 + e^{-\alpha}) + (1 - \epsilon)\log(1 + e^{\alpha}) & x \in \mathcal{X}_- \end{cases}$$

and for $\epsilon \le \frac{1}{e^{\Lambda}+1}$, the infimum $\inf_{\alpha \in A} \mathbb{E}_y[\ell(\alpha, y) \mid x]$ can be achieved by $\alpha^*(x) = \begin{cases} \Lambda & x \in \mathcal{X}_+ \\ -\Lambda & x \in \mathcal{X}_- \end{cases}$.

Therefore, by Theorem 6.2, for these distributions and loss functions, when the best-in-class classifier $h^*$ $\epsilon$-approximates $\alpha_+ = \Lambda$ over $\mathcal{X}_+$ and $\alpha_- = -\Lambda$ over $\mathcal{X}_-$, then the minimizability gap is bounded by $\epsilon$. The existence of such a hypothesis in $\mathcal{H}$ depends on the complexity of the decision surface. For example, as previously noted, when the decision surface is characterized by a hyperplane, a hypothesis set of linear functions, coupled with a sigmoid activation function, can offer a highly effective approximation (see Figure 1 for illustration).

**B. Example: multi-class classification.** Let $\epsilon \in [0, \frac{1}{2}]$. We denote by $\mathcal{X}_k$ the subset of $\mathcal{X}$ over which $\mathsf{p}(k \mid x) = 1 - \epsilon$ and $\mathsf{p}(j \mid x) = \frac{\epsilon}{n-1}$ for $j \ne k$. Let $\mathcal{H}$ be a family of functions $h$ with

$|h(x,\cdot)| \leq \Lambda$ for all $x \in \mathcal{X}$ and such that all values in $[-\Lambda, +\Lambda]$ can be reached. Thus, $A = [-\Lambda, \Lambda]^n$ for any $x \in \mathcal{X}$. Consider the multi-class logistic loss: $\ell(h, x, y) = -\log\left[\frac{e^{h(x,y)}}{\sum_{y' \in \mathcal{Y}} e^{h(x,y')}}\right]$. For any $\alpha = [\alpha^1, \ldots, \alpha^n] \in A$, we denote by $S_k = \frac{e^{\alpha^k}}{\sum_{k' \in [n]} e^{\alpha^{k'}}}$. Then, for any $x \in \mathcal{X}$ and $\alpha \in A$,

$$\mathbb{E}_y[\ell(\alpha, y) \mid x] = -(1-\epsilon)\log(S_k) - \frac{\epsilon}{n-1}\sum_{k' \neq k}\log(S_{k'}) \text{ if } x \in \mathcal{X}_k.$$

Thus, it is not hard to see that for any $\epsilon \leq \frac{n-1}{e^{2\lambda}+n-1}$, the infimum $\inf_{\alpha \in A} \mathbb{E}_y[\ell(\alpha, y) \mid x]$ can be achieved by $\alpha^*(x) = [-\Lambda, \ldots, \Lambda, \ldots, -\Lambda]$, where $\Lambda$ occupies the $k$-th position for $x \in \mathcal{X}_k$. Therefore, by Theorem 6.2, for these distributions and loss functions, when the best-in-class classifier $h^*$ $\epsilon$-approximates $\alpha_k = \left[-\Lambda, \ldots, \underset{k-\text{th}}{\Lambda}, \ldots, -\Lambda\right]$ over $\mathcal{X}_k$, then the minimizability gap is bounded by $\epsilon$. The existence of such a hypothesis in $\mathcal{H}$ depends on the complexity of the decision surface.

## K Proof for binary margin-based losses (Theorem 4.2)

**Theorem 4.2** (Upper and lower bound for binary margin-based losses). *Let $\mathcal{H}$ be a complete hypothesis set. Assume that $\Phi$ is convex, twice continuously differentiable, and satisfies the inequalities $\Phi'(0) > 0$ and $\Phi''(0) > 0$. Then, the following property holds: $\mathcal{T}(t) = \Theta(t^2)$; that is, there exist positive constants $C > 0$, $c > 0$, and $T > 0$ such that $Ct^2 \geq \mathcal{T}(t) \geq ct^2$, for all $0 < t \leq T$.*

*Proof.* Since $\Phi$ is convex and in $C^2$, $f_t$ is also convex and differentiable with respect to $u$. For any $t \in [0, 1]$, differentiate $f_t$ with respect to $u$, we have

$$f_t'(u) = \frac{1-t}{2}\Phi'(u) - \frac{1+t}{2}\Phi'(-u).$$

Consider the function $F$ defined over $\mathbb{R}^2$ by $F(t, a) = \frac{1-t}{2}\Phi'(a) - \frac{1+t}{2}\Phi'(-a)$. Observe that $F(0,0) = 0$ and that the partial derivative of $F$ with respect to $a$ at $(0,0)$ is $\Phi''(0) > 0$:

$$\frac{\partial F}{\partial a}(t, a) = \frac{1-t}{2}\Phi''(a) + \frac{1+t}{2}\Phi''(-a), \quad \frac{\partial F}{\partial a}(0,0) = \Phi''(0) > 0.$$

Consequently, by the implicit function theorem, there exists a continuously differentiable function $\overline{a}$ such that $F(t, \overline{a}(t)) = 0$ in a neighborhood $[-\epsilon, \epsilon]$ around zero. Thus, by the convexity of $f_t$ and the definition of $F$, for $t \in [0, \epsilon]$, $\inf_{u \in \mathbb{R}} f_t(u)$ is reached by $\overline{a}(t)$ and we can denote it by $a_t^*$. Then, $a_t^*$ is continuously differentiable over $[0, \epsilon]$. The minimizer $a_t^*$ satisfies the following equality:

$$f_t'(a_t^*) = \frac{1-t}{2}\Phi'(a_t^*) - \frac{1+t}{2}\Phi'(-a_t^*) = 0. \tag{16}$$

Specifically, at $t = 0$, we have $\Phi'(a_0^*) = \Phi'(-a_0^*)$. Since $\Phi$ is convex, its derivative $\Phi'$ is non-decreasing. Therefore, if $a_0^*$ were non-zero, then $\Phi'$ would be constant over the segment $[-|a_0^*|, |a_0^*|]$. This would contradict the condition $\Phi''(0) > 0$, as a constant function cannot have a positive second derivative at any point. Thus, we must have $a_0^* = 0$ and since $\Phi'$ is non-decreasing and $\Phi''(0) > 0$, we have $a_t^* > 0$ for all $t \in (0, \epsilon]$. By Theorem 4.1 and Taylor's theorem with an integral remainder, $\mathcal{T}$ can be expressed as follows: for any $t \in [0, \epsilon]$,

$$\begin{aligned}
\mathcal{T}(t) &= f_t(0) - \inf_{u \in \mathbb{R}} f_t(u) \\
&= f_t(0) - f_t(a_t^*) \\
&= f_t'(a_t^*)(0 - a_t^*) + \int_{a_t^*}^0 (0 - u)f_t''(u)\,du \qquad (f_t'(a_t^*) = 0) \\
&= \int_0^{a_t^*} u f_t''(u)\,du \\
&= \int_0^{a_t^*} u\left[\frac{1-t}{2}\Phi''(u) + \frac{1+t}{2}\Phi''(-u)\right]du. \tag{17}
\end{aligned}$$

Since $a_t^*$ is a function of class $C^1$, we can differentiate (16) with respect to $t$, which gives the following equality for any $t$ in $(0, \epsilon]$:

$$-\frac{1}{2}\Phi'(a_t^*) + \frac{1-t}{2}\Phi''(a_t^*)\frac{da_t^*}{dt}(t) - \frac{1}{2}\Phi'(-a_t^*) + \frac{1+t}{2}\Phi''(-a_t^*)\frac{da_t^*}{dt}(t) = 0.$$

Taking the limit $t \to 0$ yields

$$-\frac{1}{2}\Phi'(0) + \frac{1}{2}\Phi''(0)\frac{da_t^*}{dt}(0) - \frac{1}{2}\Phi'(0) + \frac{1}{2}\Phi''(0)\frac{da_t^*}{dt}(0) = 0.$$

This implies that

$$\frac{da_t^*}{dt}(0) = \frac{\Phi'(0)}{\Phi''(0)} > 0.$$

Since $\lim_{t \to 0} \frac{a_t^*}{t} = \frac{da_t^*}{dt}(0) = \frac{\Phi'(0)}{\Phi''(0)} > 0$, we have $a_t^* = \Theta(t)$.

Since $\Phi''(0) > 0$ and $\Phi''$ is continuous, there is a non-empty interval $[-\alpha, +\alpha]$ over which $\Phi''$ is positive. Since $a_0^* = 0$ and $a_t^*$ is continuous, there exists a sub-interval $[0, \epsilon'] \subseteq [0, \epsilon]$ over which $a_t^* \le \alpha$. Since $\Phi''$ is continuous, it admits a minimum and a maximum over any compact set and we can define $c = \min_{u \in [-\alpha, \alpha]} \Phi''(u)$ and $C = \max_{u \in [-\alpha, \alpha]} \Phi''(u)$. $c$ and $C$ are both positive since we have $\Phi''(0) > 0$. Thus, for $t$ in $[0, \epsilon']$, by (17), the following inequality holds:

$$C\frac{(a_t^*)^2}{2} = \int_0^{a_t^*} uC\, du \ge \mathcal{T}(t) = \int_0^{a_t^*} u\left[\frac{1-t}{2}\Phi''(u) + \frac{1+t}{2}\Phi''(-u)\right]du \ge \int_0^{a_t^*} uC\, du = c\frac{(a_t^*)^2}{2}.$$

This implies that $\mathcal{T}(t) = \Theta(t^2)$. $\qquad\square$

## L   Proof for comp-sum losses (Theorem 5.3)

**Theorem 5.3** (Upper and lower bound for comp-sum losses). *Assume that $\Phi$ is convex, twice continuously differentiable, and satisfies the properties $\Phi'(u) < 0$ and $\Phi''(u) > 0$ for any $u \in (0, \frac{1}{2}]$. Then, the following property holds: $\mathcal{T}(t) = \Theta(t^2)$.*

*Proof.* For any $\tau \in \left[\frac{1}{n}, \frac{1}{2}\right]$, define the function $\mathcal{T}_\tau$ by

$$\forall t \in [0, 1], \quad \mathcal{T}_\tau(t) = \sup_{|u| \le \tau}\left\{\Phi(\tau) - \frac{1-t}{2}\Phi(\tau + u) - \frac{1+t}{2}\Phi(\tau - u)\right\}$$

$$= f_{t,\tau}(0) - \inf_{|u| \le \tau} f_{t,\tau}(u),$$

where

$$f_{t,\tau}(u) = \frac{1-t}{2}\Phi_\tau(u) + \frac{1+t}{2}\Phi_\tau(-u) \quad \text{and} \quad \Phi_\tau(u) = \Phi(\tau + u).$$

We aim to establish a lower bound for $\inf_{\tau \in [\frac{1}{n}, \frac{1}{2}]} \mathcal{T}_\tau(t)$. For any fixed $\tau \in \left[\frac{1}{n}, \frac{1}{2}\right]$, this situation is parallel to that of binary classification (Theorem 4.1 and Theorem 4.2), since we have $\Phi_\tau'(0) = \Phi'(\tau) < 0$ and $\Phi_\tau''(0) = \Phi''(\tau) > 0$. Let $a_{t,\tau}^*$ denotes the minimizer of $f_{t,\tau}$ over $\mathbb{R}$. By applying Theorem 5.2 to the function $F: (t, u, \tau) \mapsto f_{t,\tau}'(u) = \frac{1-t}{2}\Phi_\tau'(u) - \frac{1+t}{2}\Phi_\tau'(-u)$ and the convexity of $f_{t,\tau}$ with respect to $u$, $a_{t,\tau}^*$ exists, is unique and is continuously differentiable over $[0, t_0'] \times \left[\frac{1}{n}, \frac{1}{2}\right]$, for some $t_0' > 0$. Moreover, by using the fact that $f_{0,\tau}'(\tau) > 0$ and $f_{0,\tau}'(-\tau) < 0$, and the convexity of $f_{0,\tau}$ with respect to $u$, we have $\left|a_{0,\tau}^*\right| \le \tau$, $\forall \tau \in \left[\frac{1}{n}, \frac{1}{2}\right]$. By the continuity of $a_{t,\tau}^*$, we have $\left|a_{t,\tau}^*\right| \le \tau$ over $[0, t_0] \times \left[\frac{1}{n}, \frac{1}{2}\right]$, for some $t_0 > 0$ and $t_0 \le t_0'$.

Next, we will leverage the proof of Theorem 4.2. Adopting a similar notation, while incorporating the $\tau$ subscript to distinguish different functions $\Phi_\tau$ and $f_{t,\tau}$, we can write

$$\forall t \in [0, t_0], \quad \mathcal{T}_\tau(t) = \int_0^{-a_{t,\tau}^*} u\left[\frac{1-t}{2}\Phi_\tau''(-u) + \frac{1+t}{2}\Phi_\tau''(u)\right]du.$$

where $a_{t,\tau}^*$ verifies

$$a_{0,\tau}^* = 0 \quad \text{and} \quad \frac{\partial a_{t,\tau}^*}{\partial t}(0) = \frac{\Phi_\tau'(0)}{\Phi_\tau''(0)} = c_\tau < 0. \tag{18}$$

We first show the lower bound $\inf_{\tau\in\left[\frac{1}{n},\frac{1}{2}\right]} -a_{t,\tau}^* = \Omega(t)$. Given the equalities (18), it follows that for any $\tau$, the following holds: $\lim_{t\to 0}\left(-a_{t,\tau}^* + c_\tau t\right) = 0$. For any $\tau \in \left[\frac{1}{n},\frac{1}{2}\right]$, $t \mapsto \left(-a_{t,\tau}^* + c_\tau t\right)$ is a continuous function over $[0,t_0]$ since $a_{t,\tau}^*$ is a function of class $C^1$. Since the infimum over a fixed compact set of a family of continuous functions is continuous, $t \mapsto \inf_{\tau\in\left[\frac{1}{n},\frac{1}{2}\right]}\left\{-a_{t,\tau}^* + c_\tau t\right\}$ is continuous. Thus, for any $\epsilon > 0$, there exists $t_1 > 0$, $t_1 \le t_0$, such that for any $t \in [0,t_1]$,

$$\left| \inf_{\tau\in\left[\frac{1}{n},\frac{1}{2}\right]}\left\{-a_{t,\tau}^* + c_\tau t\right\} \right| \le \epsilon,$$

which implies

$$\forall \tau \in \left[\frac{1}{n},\frac{1}{2}\right], \quad -a_{t,\tau}^* \ge -c_\tau t - \epsilon \ge ct - \epsilon,$$

where $c = \inf_{\tau\in\left[\frac{1}{n},\frac{1}{2}\right]} -c_\tau$. Since $\Phi_\tau'(0)$ and $\Phi_\tau''(0)$ are positive and continuous functions of $\tau$, this infimum is attained over the compact set $\left[\frac{1}{n},\frac{1}{2}\right]$, leading to $c > 0$. Since the lower bound holds uniformly over $\tau$, this shows that for $t \in [0,t_1]$, we have $\inf_{\tau\in\left[\frac{1}{n},\frac{1}{2}\right]} -a_{t,\tau}^* = \Omega(t)$.

Now, since for any $\tau \in \left[\frac{1}{n},\frac{1}{2}\right]$, $-a_{t,\tau}^*$ is a function of class $C^1$ and thus continuous, its supremum over a compact set, $\sup_{\tau\in\left[\frac{1}{n},\frac{1}{2}\right]} -a_{t,\tau}^*$, is also continuous and is bounded over $[0,t_1]$ by some $a > 0$. For $|u| \le a$ and $\tau \in \left[\frac{1}{n},\frac{1}{2}\right]$, we have $\frac{1}{2} - a \le \tau + u \le \frac{1}{2} + a$ and $\frac{1}{2} - a \le \tau - u \le \frac{1}{2} + a$. Since $\Phi''$ is positive and continuous, it reaches its minimum $C > 0$ over the compact set $\left[\frac{1}{2} - a, \frac{1}{2} + a\right]$. Thus, we can write

$$\forall t \in [0,t_1], \forall \tau \in \left[\frac{1}{n},\frac{1}{2}\right], \quad \mathfrak{T}_\tau(t) = \int_0^{-a_{t,\tau}^*} u\left[\frac{1-t}{2}\Phi_\tau''(-u) + \frac{1+t}{2}\Phi_\tau''(u)\right] du$$

$$\ge \int_0^{-a_{t,\tau}^*} u\left[\frac{1-t}{2}C + \frac{1+t}{2}C\right] du$$

$$= \int_0^{-a_{t,\tau}^*} Cu\, du = C\frac{(-a_{t,\tau}^*)^2}{2}.$$

Thus, for $t \le t_1$, we have

$$\inf_{\tau\in\left[\frac{1}{n},\frac{1}{2}\right]} \mathfrak{T}_\tau(t) \ge C\frac{(\inf_{\tau\in\left[\frac{1}{n},\frac{1}{2}\right]} -a_{t,\tau}^*)^2}{2} \ge \Omega(t^2).$$

Similarly, we aim to establish an upper bound for $\inf_{\tau\in\left[\frac{1}{n},\frac{1}{2}\right]} \mathfrak{T}_\tau(t)$. We first show the upper bound $\sup_{\tau\in\left[\frac{1}{n},\frac{1}{2}\right]} -a_{t,\tau}^* = O(t)$. Given the equalities (18), it follows that for any $\tau$, the following holds: $\lim_{t\to 0}\left(-a_{t,\tau}^* + c_\tau t\right) = 0$. For any $\tau \in \left[\frac{1}{n},\frac{1}{2}\right]$, $t \mapsto \left(-a_{t,\tau}^* + c_\tau t\right)$ is a continuous function over $[0,t_0]$ since $a_{t,\tau}^*$ is a function of class $C^1$. Since the supremum over a fixed compact set of a family of continuous functions is continuous, $t \mapsto \sup_{\tau\in\left[\frac{1}{n},\frac{1}{2}\right]}\left\{-a_{t,\tau}^* + c_\tau t\right\}$ is continuous. Thus, for any $\epsilon > 0$, there exists $t_1 > 0$, $t_1 \le t_0$, such that for any $t \in [0,t_1]$,

$$\left| \sup_{\tau\in\left[\frac{1}{n},\frac{1}{2}\right]}\left\{-a_{t,\tau}^* + c_\tau t\right\} \right| \le \epsilon,$$

which implies

$$\forall \tau \in \left[\frac{1}{n},\frac{1}{2}\right], \quad -a_{t,\tau}^* \le -c_\tau t + \epsilon \le ct + \epsilon,$$

where $c = \sup_{\tau\in\left[\frac{1}{n},\frac{1}{2}\right]} -c_\tau$. Since $\Phi_\tau'(0)$ and $\Phi_\tau''(0)$ are positive and continuous functions of $\tau$, this supremum is attained over the compact set $\left[\frac{1}{n},\frac{1}{2}\right]$, leading to $c > 0$. Since the upper bound holds uniformly over $\tau$, this shows that for $t \in [0,t_1]$, we have $\sup_{\tau\in\left[\frac{1}{n},\frac{1}{2}\right]} -a_{t,\tau}^* = O(t)$.

Now, since for any $\tau \in \left[\frac{1}{n},\frac{1}{2}\right]$, $-a_{t,\tau}^*$ is a function of class $C^1$ and thus continuous, its supremum over a compact set, $\sup_{\tau\in\left[\frac{1}{n},\frac{1}{2}\right]} -a_{t,\tau}^*$, is also continuous and is bounded over $[0,t_1]$ by some $a > 0$. For $|u| \le a$ and $\tau \in \left[\frac{1}{n},\frac{1}{2}\right]$, we have $\frac{1}{2} - a \le \tau + u \le \frac{1}{2} + a$ and $\frac{1}{2} - a \le \tau - u \le \frac{1}{2} + a$. Since $\Phi''$ is

positive and continuous, it reaches its maximum $C > 0$ over the compact set $\left[\frac{1}{2} - a, \frac{1}{2} + a\right]$. Thus, we can write

$$\forall t \in [0, t_1], \forall \tau \in \left[\frac{1}{n}, \frac{1}{2}\right], \quad \mathcal{T}_\tau(t) = \int_0^{-a_{t,\tau}^*} u\left[\frac{1-t}{2}\Phi_\tau''(-u) + \frac{1+t}{2}\Phi_\tau''(u)\right]du$$

$$\leq \int_0^{-a_{t,\tau}^*} u\left[\frac{1-t}{2}C + \frac{1+t}{2}C\right]du$$

$$= \int_0^{-a_{t,\tau}^*} Cu\,du = C\frac{(-a_{t,\tau}^*)^2}{2}.$$

Thus, for $t \leq t_1$, we have

$$\inf_{\tau \in \left[\frac{1}{n}, \frac{1}{2}\right]} \mathcal{T}_\tau(t) \leq C\frac{(\sup_{\tau \in \left[\frac{1}{n}, \frac{1}{2}\right]} -a_{t,\tau}^*)^2}{2} \leq O(t^2).$$

This completes the proof. $\qquad\square$

## M   Proof for constrained losses (Theorem 5.5)

**Theorem 5.5** (Upper and lower bound for constrained losses). *Assume that $\Phi$ is convex, twice continuously differentiable, and satisfies the properties $\Phi'(u) > 0$ and $\Phi''(u) > 0$ for any $u \geq 0$. Then, for any $A > 0$, the following property holds:*

$$\inf_{\tau \in [0,A]} \sup_{u \in \mathbb{R}} \left\{\Phi(\tau) - \frac{1-t}{2}\Phi(\tau + u) - \frac{1+t}{2}\Phi(\tau - u)\right\} = \Theta(t^2).$$

*Proof.* For any $\tau \in [0, A]$, define the function $\mathcal{T}_\tau$ by

$$\forall t \in [0, 1], \quad \mathcal{T}_\tau(t) = \sup_{u \in \mathbb{R}}\left\{\Phi(\tau) - \frac{1-t}{2}\Phi(\tau + u) - \frac{1+t}{2}\Phi(\tau - u)\right\}$$

$$= f_{t,\tau}(0) - \inf_{u \in \mathbb{R}} f_{t,\tau}(u),$$

where

$$f_{t,\tau}(u) = \frac{1-t}{2}\Phi_\tau(u) + \frac{1+t}{2}\Phi_\tau(-u) \quad\text{and}\quad \Phi_\tau(u) = \Phi(\tau + u).$$

We aim to establish a lower bound for $\inf_{\tau \in [0,A]} \mathcal{T}_\tau(t)$. For any fixed $\tau \in [0, A]$, this situation is parallel to that of binary classification (Theorem 4.1 and Theorem 4.2), since we also have $\Phi_\tau'(0) = \Phi'(\tau) > 0$ and $\Phi_\tau''(0) = \Phi''(\tau) > 0$. Let $a_{t,\tau}^*$ denotes the minimizer of $f_{t,\tau}$ over $\mathbb{R}$. By applying Theorem 5.2 to the function $F: (t, u, \tau) \mapsto f_{t,\tau}'(u) = \frac{1-t}{2}\Phi_\tau'(u) - \frac{1+t}{2}\Phi_\tau'(-u)$ and the convexity of $f_{t,\tau}$ with respect to $u$, $a_{t,\tau}^*$ exists, is unique and is continuously differentiable over $[0, t_0] \times [0, A]$, for some $t_0 > 0$.

Next, we will leverage the proof of Theorem 4.2. Adopting a similar notation, while incorporating the $\tau$ subscript to distinguish different functions $\Phi_\tau$ and $f_{t,\tau}$, we can write

$$\forall t \in [0, t_0], \quad \mathcal{T}_\tau(t) = \int_0^{a_{t,\tau}^*} u\left[\frac{1-t}{2}\Phi_\tau''(u) + \frac{1+t}{2}\Phi_\tau''(-u)\right]du.$$

where $a_{t,\tau}^*$ verifies

$$a_{0,\tau}^* = 0 \quad\text{and}\quad \frac{\partial a_{t,\tau}^*}{\partial t}(0) = \frac{\Phi_\tau'(0)}{\Phi_\tau''(0)} = c_\tau > 0. \tag{19}$$

We first show the lower bound $\inf_{\tau \in [0,A]} a_{t,\tau}^* = \Omega(t)$. Given the equalities (19), it follows that for any $\tau$, the following holds: $\lim_{t \to 0}(a_{t,\tau}^* - c_\tau t) = 0$. For any $\tau \in [0, A]$, $t \mapsto (a_{t,\tau}^* - c_\tau t)$ is a continuous function over $[0, t_0]$ since $a_{t,\tau}^*$ is a function of class $C^1$. Since the infimum over a fixed compact set of a family of continuous functions is continuous, $t \mapsto \inf_{\tau \in [0,A]}\{a_{t,\tau}^* - c_\tau t\}$ is continuous. Thus, for any $\epsilon > 0$, there exists $t_1 > 0$, $t_1 \leq t_0$, such that for any $t \in [0, t_1]$,

$$\left|\inf_{\tau \in [0,A]}\{a_{t,\tau}^* - c_\tau t\}\right| \leq \epsilon,$$

which implies
$$\forall \tau \in [0, A], \quad a^*_{t,\tau} \geq c_\tau t - \epsilon \geq ct - \epsilon,$$
where $c = \inf_{\tau \in [0,A]} c_\tau$. Since $\Phi'_\tau(0)$ and $\Phi''_\tau(0)$ are positive and continuous functions of $\tau$, this infimum is attained over the compact set $[0, A]$, leading to $c > 0$. Since the lower bound holds uniformly over $\tau$, this shows that for $t \in [0, t_1]$, we have $\inf_{\tau \in [0,A]} a^*_{t,\tau} = \Omega(t)$.

Now, since for any $\tau \in [0, A]$, $a^*_{t,\tau}$ is a function of class $C^1$ and thus continuous, its supremum over a compact set, $\sup_{\tau \in [0,A]} a^*_{t,\tau}$, is also continuous and is bounded over $[0, t_1]$ by some $a > 0$. For $|u| \leq a$ and $\tau \in [0, A]$, we have $A - a \leq \tau + u \leq A + a$ and $A - a \leq \tau - u \leq A + a$. Since $\Phi''$ is positive and continuous, it reaches its minimum $C > 0$ over the compact set $[A - a, A + a]$. Thus, we can write

$$\forall t \in [0, t_1], \forall \tau \in [0, A], \quad \mathcal{T}_\tau(t) = \int_0^{a^*_{t,\tau}} u \left[ \frac{1-t}{2} \Phi''_\tau(u) + \frac{1+t}{2} \Phi''_\tau(-u) \right] du$$
$$\geq \int_0^{a^*_{t,\tau}} u \left[ \frac{1-t}{2} C + \frac{1+t}{2} C \right] du$$
$$= \int_0^{a^*_{t,\tau}} C u \, du = C \frac{(a^*_{t,\tau})^2}{2}.$$

Thus, for $t \leq t_1$, we have

$$\inf_{\tau \in [0,A]} \mathcal{T}_\tau(t) \geq C \frac{(\inf_{\tau \in [0,A]} a^*_{t,\tau})^2}{2} \geq \Omega(t^2).$$

Similarly, we aim to establish an upper bound for $\inf_{\tau \in [0,A]} \mathcal{T}_\tau(t)$. We first show the upper bound $\sup_{\tau \in [0,A]} a^*_{t,\tau} = O(t)$. Given the equalities (19), it follows that for any $\tau$, the following holds: $\lim_{t \to 0} (a^*_{t,\tau} - c_\tau t) = 0$. For any $\tau \in [0, A]$, $t \mapsto (a^*_{t,\tau} - c_\tau t)$ is a continuous function over $[0, t_0]$ since $a^*_{t,\tau}$ is a function of class $C^1$. Since the supremum over a fixed compact set of a family of continuous functions is continuous, $t \mapsto \sup_{\tau \in [0,A]} \{a^*_{t,\tau} - c_\tau t\}$ is continuous. Thus, for any $\epsilon > 0$, there exists $t_1 > 0$, $t_1 \leq t_0$, such that for any $t \in [0, t_1]$,

$$\left| \sup_{\tau \in [0,A]} \{a^*_{t,\tau} - c_\tau t\} \right| \leq \epsilon,$$

which implies
$$\forall \tau \in [0, A], \quad a^*_{t,\tau} \leq c_\tau t + \epsilon \leq ct + \epsilon,$$
where $c = \sup_{\tau \in [0,A]} c_\tau$. Since $\Phi'_\tau(0)$ and $\Phi''_\tau(0)$ are positive and continuous functions of $\tau$, this supremum is attained over the compact set $[0, A]$, leading to $c > 0$. Since the upper bound holds uniformly over $\tau$, this shows that for $t \in [0, t_1]$, we have $\sup_{\tau \in [0,A]} a^*_{t,\tau} = O(t)$.

Now, since for any $\tau \in [0, A]$, $a^*_{t,\tau}$ is a function of class $C^1$ and thus continuous, its supremum over a compact set, $\sup_{\tau \in [0,A]} a^*_{t,\tau}$, is also continuous and is bounded over $[0, t_1]$ by some $a > 0$. For $|u| \leq a$ and $\tau \in [0, A]$, we have $A - a \leq \tau + u \leq A + a$ and $A - a \leq \tau - u \leq A + a$. Since $\Phi''$ is positive and continuous, it reaches its maximum $C > 0$ over the compact set $[A - a, A + a]$. Thus, we can write

$$\forall t \in [0, t_1], \forall \tau \in [0, A], \quad \mathcal{T}_\tau(t) = \int_0^{a^*_{t,\tau}} u \left[ \frac{1-t}{2} \Phi''_\tau(u) + \frac{1+t}{2} \Phi''_\tau(-u) \right] du$$
$$\leq \int_0^{a^*_{t,\tau}} u \left[ \frac{1-t}{2} C + \frac{1+t}{2} C \right] du$$
$$= \int_0^{a^*_{t,\tau}} C u \, du = C \frac{(a^*_{t,\tau})^2}{2}.$$

Thus, for $t \leq t_1$, we have

$$\inf_{\tau \in [0,A]} \mathcal{T}_\tau(t) \leq C \frac{(\sup_{\tau \in [0,A]} a^*_{t,\tau})^2}{2} \leq O(t^2).$$

This completes the proof. $\qquad\square$

# N  Analysis of the function of $\tau$

Let $F$ be the function defined by

$$\forall t \in \left[0, \tfrac{1}{2}\right], \tau \in \mathbb{R}, \quad F(t, \tau) = \sup_{u \in \mathbb{R}} \left\{ \Phi(\tau) - \frac{1-t}{2}\Phi(\tau + u) - \frac{1+t}{2}\Phi(\tau - u) \right\},$$

where $\Phi$ is a convex function in $C^2$ with $\Phi', \Phi'' > 0$. In light of the analysis of the previous sections, for any $(\tau, t)$, there exists a unique function $a_{t,\tau}$ solution of the maximization (supremum in $F$), a $C^1$ function over a neighborhood $U$ of $(\tau, 0)$ with $a_{0,\tau} = 0$, $a_{t,\tau} > 0$ for $t > 0$, and $\frac{\partial a_{t,\tau}}{\partial t}(0, \tau) = \frac{\Phi'(\tau)}{\Phi''(\tau)} = c_\tau$. Thus, we have $\lim_{t \to 0} \frac{a_{t,\tau}}{tc_\tau} = 1$. The optimality of $a_{t,\tau}$ implies

$$\frac{1-t}{2}\Phi'(\tau + a_{t,\tau}) = \frac{1+t}{2}\Phi'(\tau - a_{t,\tau}).$$

Thus, the partial derivative of $F$ over the appropriate neighborhood $U$ is given by

$$
\begin{aligned}
\frac{\partial F}{\partial \tau}(t, \tau) &= \Phi'(\tau) - \frac{1-t}{2}\Phi'(\tau + a_{t,\tau})\left(\frac{\partial a_{t,\tau}}{\partial \tau}(t, \tau) + 1\right) - \frac{1+t}{2}\Phi'(\tau - a_{t,\tau})\left(-\frac{\partial a_{t,\tau}}{\partial \tau}(t, \tau) + 1\right) \\
&= \Phi'(\tau) - \frac{1-t}{2}\Phi'(\tau + a_{t,\tau})\left(\frac{\partial a_{t,\tau}}{\partial \tau}(t, \tau) + 1 - \frac{\partial a_{t,\tau}}{\partial \tau}(t, \tau) + 1\right) \\
&= \Phi'(\tau) - (1-t)\Phi'(\tau + a_{t,\tau}).
\end{aligned}
$$

Since $\Phi'$ is continuous, by the mean value theorem, there exists $\xi \in (\tau, \tau + a_{t,\tau})$ such that $\Phi'(\tau + a_{t,\tau}) - \Phi'(\tau) = a_{t,\tau}\Phi''(\xi)$. Thus, we can write

$$
\begin{aligned}
\frac{\partial F}{\partial \tau}(t, \tau) &= \Phi'(\tau) - (1-t)\Phi'(\tau) - (1-t)a_{t,\tau}\Phi''(\xi) \\
&= t\Phi'(\tau) - (1-t)a_{t,\tau}\Phi''(\xi) \\
&= t\Phi'(\tau)\left[1 - (1-t)\frac{a_{t,\tau}}{tc_\tau}\frac{\Phi''(\xi)}{\Phi''(\tau)}\right].
\end{aligned}
$$

Note that if $\Phi''$ is locally non-increasing, then we have $\Phi''(\xi) \leq \Phi''(\tau)$ and for $t$ sufficiently small, since $\Phi'$ is increasing and $\frac{a_{t,\tau}}{tc_\tau} \sim 1$:

$$\frac{\partial F}{\partial \tau}(t, \tau) \geq t\Phi'(\tau)\left[1 - (1-t)\frac{a_{t,\tau}}{tc_\tau}\right] \geq 0. \tag{20}$$

In that case, for any $A > 0$, we can find a neighborhood $\mathcal{O}$ of $t$ around zero over which $\frac{\partial F}{\partial \tau}(t, \tau)$ is defined for all $(t, \tau) \in \mathcal{O} \times [0, A]$ and $\frac{\partial F}{\partial \tau}(t, \tau) \geq 0$. From this, we can conclude that the infimum of $F$ over $\tau \in [0, A]$ is reached at zero for $t$ sufficiently small ($t \in \mathcal{O}$).

# O  Generalization bounds

Let $S = ((x_1, y_1), \ldots, (x_m, y_m))$ be a sample drawn from $\mathcal{D}^m$. Denote by $\widehat{h}_S$ an empirical minimizer within $\mathcal{H}$ for the surrogate loss $\ell$: $\widehat{h}_S \in \operatorname{argmin}_{h \in \mathcal{H}} \frac{1}{m}\sum_{i=1}^m \ell(h, x_i, y_i)$. Let $\mathcal{H}_\ell$ denote the hypothesis set $\{(x, y) \mapsto \ell(h, x, y) : h \in \mathcal{H}\}$ and $\mathfrak{R}_m^\ell(\mathcal{H})$ its Rademacher complexity. We also write $B_\ell$ to denote an upper bound for $\ell$. Then, given the following $\mathcal{H}$-consistency bound:

$$\forall h \in \mathcal{H}, \quad \mathcal{E}_{\ell_{0-1}}(h) - \mathcal{E}_{\ell_{0-1}}^*(\mathcal{H}) + \mathcal{M}_{\ell_{0-1}}(\mathcal{H}) \leq \Gamma(\mathcal{E}_\ell(h) - \mathcal{E}_\ell^*(\mathcal{H}) + \mathcal{M}_\ell(\mathcal{H})), \tag{21}$$

for any $\delta > 0$, with probability at least $1 - \delta$ over the draw of an i.i.d. sample $S$ of size $m$, the following estimation bound holds for $\widehat{h}_S$:

$$\forall h \in \mathcal{H}, \quad \mathcal{E}_{\ell_{0-1}}(h) - \mathcal{E}_{\ell_{0-1}}^*(\mathcal{H}) \leq \Gamma\left(4\mathfrak{R}_m^{\mathsf{L}}(\mathcal{H}) + 2B_{\mathsf{L}}\sqrt{\frac{\log\frac{2}{\delta}}{2m}} + \mathcal{M}_\ell(\mathcal{H})\right) - \mathcal{M}_{\ell_{0-1}}(\mathcal{H}).$$

*Proof.* By the standard Rademacher complexity bounds [Mohri et al., 2018], for any $\delta > 0$, with probability at least $1 - \delta$, the following holds for all $h \in \mathcal{H}$:

$$\left|\mathcal{E}_\ell(h) - \widehat{\mathcal{E}}_{\ell,S}(h)\right| \leq 2\mathfrak{R}_m^\ell(\mathcal{H}) + B_\ell\sqrt{\frac{\log(2/\delta)}{2m}}.$$

For any $\epsilon > 0$, by definition of the infimum, there exists $h^* \in \mathcal{H}$ such that $\mathcal{E}_\ell(h^*) \le \mathcal{E}_\ell^*(\mathcal{H}) + \epsilon$. By the definition of $\widehat{h}_S$, we obtain

$$
\begin{aligned}
\mathcal{E}_\ell(\widehat{h}_S) - \mathcal{E}_\ell^*(\mathcal{H}) &= \mathcal{E}_\ell(\widehat{h}_S) - \widehat{\mathcal{E}}_{\ell,S}(\widehat{h}_S) + \widehat{\mathcal{E}}_{\ell,S}(\widehat{h}_S) - \mathcal{E}_\ell^*(\mathcal{H}) \\
&\le \mathcal{E}_\ell(\widehat{h}_S) - \widehat{\mathcal{E}}_{\ell,S}(\widehat{h}_S) + \widehat{\mathcal{E}}_{\ell,S}(h^*) - \mathcal{E}_\ell^*(\mathcal{H}) \\
&\le \mathcal{E}_\ell(\widehat{h}_S) - \widehat{\mathcal{E}}_{\ell,S}(\widehat{h}_S) + \widehat{\mathcal{E}}_{\ell,S}(h^*) - \mathcal{E}_\ell^*(h^*) + \epsilon \\
&\le 2\left[2\mathfrak{R}_m^\ell(\mathcal{H}) + B_\ell\sqrt{\tfrac{\log(2/\delta)}{2m}}\right] + \epsilon.
\end{aligned}
$$

Since the inequality holds for all $\epsilon > 0$, it implies the following:

$$
\mathcal{E}_\ell(\widehat{h}_S) - \mathcal{E}_\ell^*(\mathcal{H}) \le 4\mathfrak{R}_m^\ell(\mathcal{H}) + 2B_\ell\sqrt{\tfrac{\log(2/\delta)}{2m}}.
$$

Plugging in this inequality in the $\mathcal{H}$-consistency bound (21) completes the proof. $\qquad\square$

These bounds for surrogate loss minimizers, expressed in terms of minimizability gaps, offer more detailed and informative insights compared to existing bounds based solely on approximation errors. Our analysis of growth rates suggests that for commonly used smooth loss functions, $\Gamma$ varies near zero with a square-root dependency. Furthermore, this dependency cannot be generally improved for arbitrary distributions.

## P  Future work

We demonstrated a universal square-root growth rate for smooth surrogate losses commonly used in binary and multi-class classification. This result holds across all data distributions. A promising direction for future research would be to further investigate how incorporating specific distributional assumptions could refine these results.

