# OpenReview forum: "A Universal Growth Rate for Learning with Smooth Surrogate Losses"
_NeurIPS.cc/2024/Conference — NeurIPS 2024 poster_

### Official Review · Reviewer_jALT · 2024-07-08

**Soundness:** 4
**Presentation:** 3
**Contribution:** 4
**Rating:** 8
**Confidence:** 4

**Summary:**

This paper analyzes the growth rate of the H-consistency bounds (which subsume excess risk bounds) for various smooth surrogate losses commonly used in binary and multiclass classification. Specifically, for binary classification, the work establishes a tight square-root growth rate near zero (under mild conditions) for margin-based surrogate losses. For multiclass classification, the work establishes a tight square-root growth rate near zero (under mild conditions) for two families of surrogate losses: comp-sum and constrained losses. Finally, the work also studies how the number of classes affects these bounds, as well as the minimizability gaps in the bounds.

**Strengths:**

**Originality**

A comprehensive analysis of the growth rate of the H-consistency bounds (which subsume excess risk bounds) for various smooth surrogate losses commonly used in binary and multiclass classification:
-  For binary classification, the work establishes a tight square-root growth rate near zero (under mild conditions) for margin-based surrogate losses (Theorem 4.2). In particular, the lower bound requires weaker conditions than [Frongillo and Waggoner, 2021,
Theorem 4], and the upper bound is new.
- For multiclass classification, the work establishes a tight square-root growth rate near zero (under mild conditions) for two families of surrogate losses: comp-sum and constrained losses (Theorems 5.3 and 5.5).

Related work has been properly cited.

**Quality**

The work is technically sound. Proofs are given for theoretical results.

**Clarity**

The paper is clearly written and well-organized. Although it is a bit dry, as a researcher working in the related field, I did not find it very hard to read.

**Significance**

The comprehensive analysis presented in this work promotes a deeper understanding of different surrogate losses. It is helpful for researchers studying surrogate losses and consistency.

**Weaknesses:**

I did not find any obvious weaknesses.

**Questions:**

1. Based on this work, do you have any concrete suggestions for practitioners to choose surrogate losses (assuming they do not know about H-consistency at all)?

**Limitations:**

The authors have adequately addressed the limitations.

---

> ### Author Rebuttal · Authors · 2024-08-06
>
> Thank you for your appreciation of our work. Please find our detailed responses below.
>
> **Questions: Based on this work, do you have any concrete suggestions for practitioners to choose surrogate losses (assuming they do not know about H-consistency at all)?**
>
> **Response:** We have discussed some practical implications of our analysis in Section 6 and Appendix H.
>
> In practice, in realizable or nearly realizable cases where minimizability gaps are zero or relatively small, we believe that the logistic loss is a favorable choice because it is advantageous for optimization and its bound is independent of the number of classes. This in fact can partly explain its widespread practical application.
>
> In other scenarios, both the number of classes and the minimizability gaps are essential in loss selection and a good choice out of the family of comp-sum losses might suggest a parameter $\tau$ closer to 2.
>
> More generally, our theoretical analysis can help in selecting surrogate loss functions by considering several factors related to their $H$-consistency bounds:
> - Their growth rate. For example, the growth rate for polyhedral surrogate losses is linear, while it is square-root for smooth losses.
> - Their optimization property. For example, smooth losses are more favorable for optimization compared to polyhedral losses, particularly with deep neural networks.
> - Their functional form. For example, comp-sum loss functions can vary in their forms of $H$-consistency bounds due to the dependency on the number of classes (see Section 6).
> - Their approximation property. For example, the minimizability gaps differ in $H$-consistency bounds for smooth surrogate losses, even with the same growth rate (see comparison in Section 6). Also, under certain conditions, minimizability gaps can be zero or close to zero.
>
> For smooth loss functions, in particular comp-sum losses, please see further our discussion in Section 6.2.

---

> > ### Comment · Reviewer_jALT · 2024-08-09
> > **Increase my rating to 8**
> >
> > Thank you for the responses. I think this work is solid and of significant value to the learning theory community. I want to increase my rating to 8, reflecting its significance.

---

> > > ### Author Response · Authors · 2024-08-11
> > >
> > > We would like to express our gratitude to the reviewer for their positive feedback and for recognizing the significance of our work.

---

### Official Review · Reviewer_s9tD · 2024-07-09

**Soundness:** 3
**Presentation:** 3
**Contribution:** 3
**Rating:** 7
**Confidence:** 3

**Summary:**

Since optimizing zero-one loss is intractable and it does not have properties such as differentiability, a common approach in learning theory is to replace it with a surrogate loss function. H-consistency bounds relate the excess error for surrogate loss to zero-one loss. This paper establishes a square-root growth rate near zero for smooth surrogate losses in binary and multi-class classification, providing both upper and lower bounds under mild assumptions.

**Strengths:**

The paper proves both upper and lower bounds for H-consistency with smooth surrogate losses. Previous results only provided lower bounds, but this paper presents lower bounds with fewer conditions and also shows a matching upper bound of the square root, applicable to both binary classification and multiclass classification, such as Comp-sum losses.

**Weaknesses:**

The paper studies only smooth surrogate losses, not piecewise linear ones such as hinge loss. However, using smooth surrogate loss functions is very common in machine learning applications.

**Questions:**

The results are based on the assumption that the hypothesis class is complete. Can you explain why this condition is necessary and what happens to the growth rate when this is not satisfied? It seems that many practical hypothesis classes don't meet this condition.

**Limitations:**

The paper does not have any potential negative societal impact.

---

> ### Author Rebuttal · Authors · 2024-08-06
>
> Thank you for your appreciation of our work. Please find our detailed responses below.
>
> **Weaknesses: The paper studies only smooth surrogate losses, not piecewise linear ones such as hinge loss. However, using smooth surrogate loss functions is very common in machine learning applications.**
>
> **Response:** Indeed, smooth surrogate loss functions are the most commonly used ones in current training of neural networks. Our focus on smooth surrogate loss functions was also motivated by the fact that the prior work of Frongillo and Waggoner (2021) implies a linear growth rate for polyhedral losses. In Appendix G, we give a brief discussion comparing polyhedral losses with smooth losses.
>
> **Questions: The results are based on the assumption that the hypothesis class is complete. Can you explain why this condition is necessary and what happens to the growth rate when this is not satisfied? It seems that many practical hypothesis classes don't meet this condition.**
>
> **Response:** Completeness means that for any instance, the set of possible scores generated by a hypothesis set spans $\mathbb{R}$. This condition is met by common hypothesis sets, for instance the family of all linear models, $H_{\mathrm{lin}} = \\{ (x,y) \mapsto W_{y} x + b_{y} \\}$, or that of multi-layer neural networks, $H_{\mathrm{NN}} = \\{ (x, y)\mapsto u_y \cdot \rho_{n}(W_{y, n}(\cdots \rho_2(W_{y, 2} \rho_1(W_{y, 1} x + b_{y, 1})+b_{y, 2})\cdots)+b_{y, n}) \\}$, where $\rho_j$ is an activation function.
>
> More broadly, our analysis and results can be extended to bounded hypothesis sets even when completeness is not satisfied. This can be done by leveraging the characterization of the error transformation function given by Awasthi et al. [2022b] and Mao et al. [2023b] without assuming completeness.
>
> We expect that the square root growth rate still holds in bounded cases for smooth surrogate losses. This is because, for bounded hypothesis sets, the estimation error transformation function typically admits two segments; however, our primary focus is on the segment where $t$ is close to zero, which aligns with the complete case. Thus, a similar analysis can be applied in these situations as well. We will elaborate on this in the final version.

---

### Official Review · Reviewer_cQ3k · 2024-07-12

**Soundness:** 3
**Presentation:** 3
**Contribution:** 3
**Rating:** 7
**Confidence:** 2

**Summary:**

This paper presents a comprehensive analysis of the growth rate of H-consistency bounds (and excess error bounds) for various
surrogate losses for some intractable loss used in classification. The authors prove a square-root growth rate near zero for smooth margin-based surrogate losses in binary classification and comp-sum and constrained loss used in multi-class classification, providing both upper and lower bounds under mild assumptions.

**Strengths:**

The paper provides solid analysis for the H-consistency and excess error bound and these provide good guidance for selecting good surrogate loss for classification tasks where the target loss is hard to optimize.
The theoretical analysis is novel and provides good insight for handling intractable target loss.

**Weaknesses:**

The paper should provide a more intuitive statement on the motivation and implication of these error bounds and provide more insights about the proof.

**Questions:**

1. The error bounds are local bounds. Can the author provide insights on how to have global bounds and the neighborhood for the local bounds to hold?
 2. Can the authors provide more insights on how these bounds help us select surrogate functions?

---

> ### Author Rebuttal · Authors · 2024-08-06
>
> Thank you for your appreciation of our work and for your suggestions to improve its readability. Please find our detailed responses below.
>
> **Weaknesses: The paper should provide a more intuitive statement on the motivation and implication of these error bounds and provide more insights about the proof.**
>
> **Response:** Thank you for your suggestions. We will continue to work on improving our presentation for readers. The addition of one extra page in the final version will also allow us to include more detailed discussions of the motivation, implications, and proof techniques in the main body.
>
> **Questions:**
>
> **1. The error bounds are local bounds. Can the author provide insights on how to have global bounds and the neighborhood for the local bounds to hold?**
>
> **Response:** $H$-consistency bounds hold for all predictors $h$ in the hypothesis set $H$ considered and for all distributions. We provide a tight analysis of their growth rate for smooth surrogate losses in both binary and multi-class classification, demonstrating that it follows a square-root function.
>
> We suspect the reviewer refers to these square-root growth rate bounds as "local bounds" and is inquiring about "global bounds" that do not rely on growth rates. Previous works by Awasthi et al. (2022a,b) and Mao et al. (2023b,e) offer precise global bounds for both binary and multi-class classification across various commonly used loss functions, along with general methods for deriving such bounds. Our paper focuses on analyzing the growth rate of any of these bounds for smooth loss functions.
>
> Please let us know if we are not answering the question appropriately, as we are unsure about what meant. Thank you.
>
> **2. Can the authors provide more insights on how these bounds help us select surrogate functions?**
>
> **Response:** Our theoretical analysis assists in selecting surrogate loss functions by considering several factors related to their $H$-consistency bounds:
>
> - Their growth rate. For example, the growth rate for polyhedral surrogate losses is linear, while it is square-root for smooth losses.
>
> - Their optimization property. For example, smooth losses are more favorable for optimization compared to polyhedral losses, particularly with deep neural networks.
>
> - Their functional form. For example, comp-sum loss functions can vary in their forms of $H$-consistency bounds due to the dependency on the number of classes (see Section 6).
>
> - Their approximation property. For example, the minimizability gaps differ in $H$-consistency bounds for smooth surrogate losses, even with the same growth rate (see comparison in Section 6). Also, under certain conditions, minimizability gaps can be zero or close to zero.
>
> For smooth loss functions, in particular comp-sum losses, please see further our discussion in Section 6.2.

---

> > ### Comment · Reviewer_cQ3k · 2024-08-13
> >
> > Thanks for the response. I will keep my score.

---

> > > ### Author Response · Authors · 2024-08-13
> > >
> > > Thank you for your comments. We appreciate the reviewer's valuable suggestions and feedback.

---

### Official Review · Reviewer_ZEpV · 2024-07-22

**Soundness:** 3
**Presentation:** 3
**Contribution:** 3
**Rating:** 5
**Confidence:** 2

**Summary:**

The paper provides an analysis of the growth rate of H-consistency bound for surrogate losses in binary and multi-class classification. The authors prove square root growth rate near zero for smooth margin-based surrogate losses for binary classification as well as  for smooth comp-sum and constrained losses for multiclass classification. Since minimizability gaps makes a big difference between the bounds of different surrogates, the authors also analyze these gaps to guide in selecting better surrogates.  In section , the authors introduce H consistency bounds and they build upon this to derive results for binary and multi-class classification in later sections.

**Strengths:**

I must say, I am not an expert in this area of research. I find this paper pretty well written. In theorem 4.2, the authors prove that the transformation function tau is precisely of the order t^2 for a class of margin based loss function that is smooth and follows some other properties as mentioned in the main statement.  Hence, the growth rate for these loss functions is precisely square-root.  Further, the authors derive similar result for multiclass classification for comp sum losses and constrained losses. Because there is a minimizability gap term in the H consistency bound hence even with identical growth rates, surrogate losses can vary in their H-consistency bounds. The authors show that in the case of multiclass classification, minimizability gap scales with number of classes.

**Weaknesses:**

I have a few very basic questions, regarding the work.
1. I understand that H consistency-based bound helps convert a surrogate-based bound to a bound that we require. How is it an improvement over earlier work Bartlett et al. (Convexity, Classification, and Risk Bounds). H consistence based bound contains the term minimizability gap which was not existent in the previous work that I cited.


2. I also understand that the faster the bound for surrogate loss will be, the better bound for the actual loss could be obtained. However, I am wondering if we provide a fast rate using small ball method or local Rademacher complexity based method for the surrogates, under what conditions the fast rate for the actual loss can still be recovered or is it lost always ?

It would be also great if you could explain simply the implications of getting a precise rate for the transformation function as I understand it might not give you a lower bound on the estimation error because of equation 2 is not an eqaulity. Am I missing something?

I am currently giving it a borderline accept and am happy to reconsider it after the rebuttal.

**Questions:**

See above.

**Limitations:**

See above.

---

> ### Author Rebuttal · Authors · 2024-08-06
>
> Thank you for your encouraging review. We have carefully addressed all the questions raised. Please find our detailed responses below and let us know if there is any other question.
>
> **1. I understand that H consistency-based bound helps convert a surrogate-based bound to a bound that we require. How is it an improvement over earlier work Bartlett et al. (Convexity, Classification, and Risk Bounds). H consistency based bound contains the term minimizability gap which was not existent in the previous work that I cited.**
>
> **Response:** The results of Bartlett et al. (2006) hold only for the family of all measurable cases, that is for $H = H_{\mathrm{all}}$. They also focus exclusively on binary classification (other studies, such as those of Tewari and Bartlett (2007) and Zhang (2004) analyze certain multi-class classification loss functions).
>
> In contrast, $H$-consistency bounds hold for arbitrary hypothesis sets. They provide the tightest possible upper bound on the estimation error for the actual loss, such as the zero-one loss, in terms of the surrogate estimation error, for an arbitrary hypothesis set $H$. They admit as special cases the excess bounds of Bartlett et al. (2006) in binary classification, when setting $H$ to $H_{\mathrm{all}}$.
>
> For the more realistic scenario where the hypothesis set $H$ does not include all measurable functions, as demonstrated in Appendix C, under general assumptions, minimizability gap terms appear. These are necessary terms to relate the surrogate estimation error to the actual estimation error. In the special case $H = H_{\mathrm{all}}$, minimizability gaps vanish. They are also zero in realizable cases.
>
> Thus, $H$-consistency bounds provide a strict generalization of previous work, needed to analyze the standard case where $H$ does not include all measurable functions. Our paper further provides a detailed analysis of minimizability gaps.
>
> **2. I also understand that the faster the bound for surrogate loss will be, the better bound for the actual loss could be obtained. However, I am wondering if we provide a fast rate using small ball method or local Rademacher complexity based method for the surrogates, under what conditions the fast rate for the actual loss can still be recovered or is it lost always?**
>
> **Response:** This is a very good question. It is well-established that local Rademacher complexity or small ball methods can be used to derive fast rate generalization bounds under Tsybakov noise conditions. Remarkably, under the same conditions, indeed, one can also derive more favorable $H$-consistency bounds (as shown in our concurrent work), that is with better exponents. Combining both yields more favorable generalization bounds on the actual loss, as suspected by the Reviewer. We will elaborate on this in the final version.
>
> More generally, more favorable $H$-consistency bounds can be derived under various distributional assumptions.
>
> **3. It would be also great if you could explain simply the implications of getting a precise rate for the transformation function as I understand it might not give you a lower bound on the estimation error because of equation 2 is not an equality. Am I missing something?**
>
> **Response:** Like excess error bounds, $H$-consistency bounds are worst-case bounds that hold for any distribution. As noted in Lines 232-235, estimation error transformation functions provide tight $H$-consistency bounds. This means that for any $t \in [0, 1]$, there exists a hypothesis $h \in H$ and a distribution such that the inequality of the bound can be achieved. Thus, a square root growth rate for transformation functions implies a square root growth rate of estimation errors in the worst case.

---

> > ### Author Response · Authors · 2024-08-13
> >
> > Dear reviewer, the deadline for the end of the discussion period is approaching. We wanted to confirm that we addressed all your questions suitably. If there other questions we could address, please let us know. Thank you.

---

### Decision · Program_Chairs · 2024-09-25

**Decision:**

Accept (poster)

**Comment:**

This paper further develops the theory of H-consistency, which relates the effect of minimizing a surrogate loss
over a hypothesis set H to the zero-one loss. In particular, one contribution of this work is to prove relatively general
results to determine the rate in the case of smooth loss functions. This paper builds upon the discoveries of previous works,
but all of the reviewers appreciated the current contribution at a technical level and discussion of the position of this work to previous work. I recommend acceptance but encourage the authors to revise the final version of the work based on these discussions.